# Chain of Execution Supervision Promotes General Reasoning in Large Language Models

**Nuo Chen**♠   **Zehua Li**♣   **Keqin Bao**◇   **Junyang Lin**♠*   **Dayiheng Liu**♠*

♠Qwen Team, Alibaba
♣Hong Kong University of Science and Technology (Guangzhou)
◇University of Science and Technology of China
chennuo26@gmail.com

## Abstract

Building robust and general reasoning ability is a central goal in the development of large language models (LLMs). Recent efforts increasingly turn to code as a rich training source, given its inherent logical structure and diverse reasoning paradigms—such as divide-and-conquer, topological ordering, and enumeration. However, reasoning in code is often expressed implicitly and entangled with syntactic or implementation noise, making direct training on raw code suboptimal. To address this, we introduce TracePile, a large-scale corpus of 2.6 million samples that transforms code execution into explicit, step-by-step chain-of-thought style rationales, which we call Chain of Execution (CoE). The corpus spans domains including mathematics, classical algorithms and algorithmic competition, and is enriched with variable-tracing questions and code rewritings to enhance logical granularity and code diversity. We evaluate TracePile using three training setups—continue-pretraining, instruction tuning after pretraining, and two-stage finetuning. Experiments across four base models (LLaMA 3, LLaMA 3.1, Qwen-2.5, and Qwen-2.5 Coder) and 20 benchmarks covering math, code, logic, and algorithms demonstrate consistent improvements. Notably, TracePile boosts LLaMA3.1-8B by 7.1% on average across nine math datasets and delivers clear gains on LiveCodeBench, CRUX, and MMLU under two-stage finetuning.

## 1 Introduction

Reasoning ability—a fundamental and foundational competence of large language models (LLMs)—is a key indicator determining whether these models can effectively progress toward artificial general intelligence (AGI). While LLMs have shown remarkable success in language understanding and generation, it is their capacity for complex reasoning—such as mathematical reasoning and code generation—that remains the most critical and challenging benchmark [24, 40, 53, 7, 56, 49, 47, 42]. Recent works have increasingly focused on boosting these reasoning abilities, with one of the most widely adopted approaches being the large-scale collection or synthesis of high-quality, task-specific reasoning data. Training on large volumes of carefully curated or synthetically generated reasoning data systematically strengthens the model's capacity for mathematical reasoning, code generation, and multi-step problem-solving [39, 2, 52, 50, 48, 17, 31]. However, despite these advances, current data-driven methods primarily enhance performance within specific reasoning domains rather than achieving broad, generalizable reasoning capabilities. Developing a corpus that incorporates diverse reasoning patterns and paradigms represents a promising and important direction for advancing models toward more robust and transferable reasoning abilities.

---

*Corresponding author.

39th Conference on Neural Information Processing Systems (NeurIPS 2025).

Code corpora hold significant potential for advancing general reasoning abilities, as they inherently encapsulate a diverse range of logical problem-solving techniques and reasoning patterns [45, 22, 27]. Mathematical code often encodes rich mathematical properties and symbolic operations, while classical algorithmic code—for example, in graph theory—embeds essential algorithmic ideas such as divide-and-conquer (e.g., merge sort), topological reasoning (e.g., topological sort in directed acyclic graphs), and enumeration strategies (e.g., combinatorial generation or backtracking) [44, 34]. However, simply training models directly on raw code is suboptimal because the relevant reasoning signals are often implicit, buried within syntactic details, or entangled with noisy implementation artifacts, making it difficult for models to extract the underlying reasoning patterns effectively. To address this, CodeI/O [27] has sought to transform raw code files into structured code functions paired with corresponding questions, enabling the model to treat the code as a reference when generating chain-of-thought (CoT) solutions. Yet, such datasets still overlook fine-grained information embedded in the execution process itself. We argue that converting each step of code execution into a CoT-style natural language narration would produce data that is not only logically rigorous and well-structured but also offers clearer, more interpretable reasoning traces. Training LLMs in such a step-by-step code execution corpus has the potential to enhance their general reasoning capabilities significantly.

To this end, we construct a large-scale step-by-step code execution corpus, which we name **TracePile** —a richly diverse dataset designed to capture fine-grained diverse reasoning signals across a variety of domains. We refer to this CoT-style step-by-step code execution as **Chain of Execution** (**CoE**). Specifically, we collect a broad range of queries and associated code snippets from multiple sources, including mathematical problems, algorithmic competition datasets, and classical algorithms. Frontier LLMs are prompted in a few-shot setting to generate detailed CoE solutions that follow the execution trace of the code, transforming each computational step into a structured and interpretable reasoning narration. To further enrich the corpus and improve logical granularity, we introduce two key augmentation strategies. First, we generate additional fine-grained questions for certain algorithmic functions, such as asking the model to trace the state changes of specific variables throughout execution. These targeted questions lead to more localized and logically dense CoE narrations, helping models internalize detailed procedural reasoning. Second, we apply systematic code rewritings to introduce variants—for instance, through structural refactoring or alternate algorithm implementations. This not only diversifies the syntactic surface of the training data but also promotes the model's robustness in understanding semantically equivalent yet stylistically different codes, encouraging broader generalization across reasoning tasks.

As a result, we collect 2.6 million high-quality CoE samples in TracePile. To comprehensively evaluate the utility of this corpus, we design three experimental settings: 1) We perform continue-pretraining on TracePile, serving as an intermediate step to enhance the reasoning abilities of the base model by exposing it to fine-grained, step-by-step execution patterns. 2) We apply general instruction tuning after continue-pretraining, allowing the model to retain broad instruction-following abilities while leveraging the specialized reasoning improvements gained from TracePile. 3) We adopt a two-stage finetuning approach, where the model is first tuned on TracePile and then on general instruction datasets, serving as a structured adaptation path that integrates specialized reasoning gains while maintaining strong general alignment across diverse tasks.

We demonstrate the effectiveness of TracePile across four different base models: LLaMA 3 & 3.1 [15], Qwen-2.5 [38], and Qwen-2.5 Coder [21]. We conduct comprehensive evaluations on as many as **20** datasets, covering four major reasoning domains: **mathematical reasoning**, **coding**, **logical reasoning**, and **algorithmic problem solving**. Across these benchmarks, models trained with TracePile consistently achieve superior performance. For example, after continue-pretraining, TracePile boosts LLaMA3.1-8B-Base by an average of 7.1% improvement across nine mathematical reasoning datasets. Moreover, following two-stage finetuning, models show clear gains on datasets such as LiveCodeBench [22], CRUX [16], and Zebra Logic [29], demonstrating TracePile's ability to enhance both domain-specific and general reasoning capabilities.

## 2   TracePile

In this section, we introduce TracePile. We begin with collecting data sources and extracting raw queries and code corpus. Then we enrich the data diversity with query and code diversification. Subsequently, we review the process of obtaining the chain of execution data and filtering strategies. Finally, the statistics of TracePile are presented. Figure 1 shows the overview of TracePile.

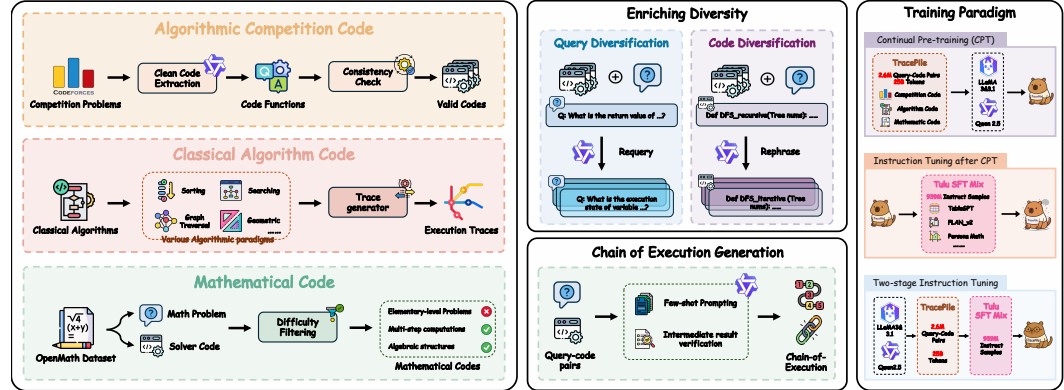

Figure 1: The curation process of TracePile and training pipelines. The left part indicates the sources of Tracepile include algorithmic competition, classical algorithm and mathematical code. The middle part shows the strategies to enrich the data diversity and CoE-style data generation. The right part includes the three different training paradigm in this work.

## 2.1 Data Sources

To build TracePile, we curate step-by-step execution data from three primary code sources, each designed to capture diverse and fine-grained reasoning patterns.

**Algorithmic Competition Code.** We collect code from algorithmic competition problems, primarily using open-source Codeforces submissions [2]. However, raw competition data often contains noisy, monolithic scripts with unclear function boundaries and hardcoded I/O. To address this, we leverage Qwen-2.5-72B-Instruct to accurately extract clean code functions along with corresponding input-output pairs (Prompts are in Appendix A). We further validate correctness by executing the code and checking whether the generated output matches the extracted reference output—only retaining samples that pass this consistency check. This ensures both syntactic validity and semantic fidelity.

**Classical Algorithm Code.** We include classical algorithm implementations, inspired by prior works such as CLRS [34] and GraphInstruct [32]. We incorporate 30 classical algorithms drawn from Cormen [10], spanning a broad spectrum of algorithmic paradigms, including sorting, searching, divide-and-conquer, greedy strategies, dynamic programming, graph traversal, string manipulation, and geometric computations. Using the official CLRS-text generator, we synthesize high-quality inputs, outputs, and intermediate execution traces. For graph theory-related problems, we utilize official generator from [32]. However, such traces from the above generator are typically presented as symbolic states or graphs rather than natural, interpretable CoT-style narratives. They will be used to check CoE's intermediate results.

**Mathematical Code.** We incorporate mathematical code, which often involves multi-step decomposition, symbolic manipulation, and numerical reasoning—making it ideal for training models on compositional thinking. We adopt OpenMath [43], a large-scale synthetic dataset of math problems paired with solver code. To increase the difficulty and reasoning richness, we apply a model-based filtering strategy: for each problem, we sample three independent responses using LLaMA3-8B. If the model answers correctly in all three attempts, we consider the problem too simple and discard it.

## 2.2 Enriching Diversity

While TracePile covers a broad range of code sources, the original code and query distributions still exhibit notable limitations. For instance, classical algorithm implementations often follow highly standardized coding styles, such as fixed loop structures, hardcoded base cases, or commonly reused recursion patterns. This lack of structural diversity can limit the variety of reasoning challenges presented to the model. To further improve the reasoning generalization potential of the dataset, we introduce two augmentation strategies aimed at enriching both the query and code.

---

[2]https://huggingface.co/datasets/MatrixStudio/Codeforces-Python-Submissions?row=0

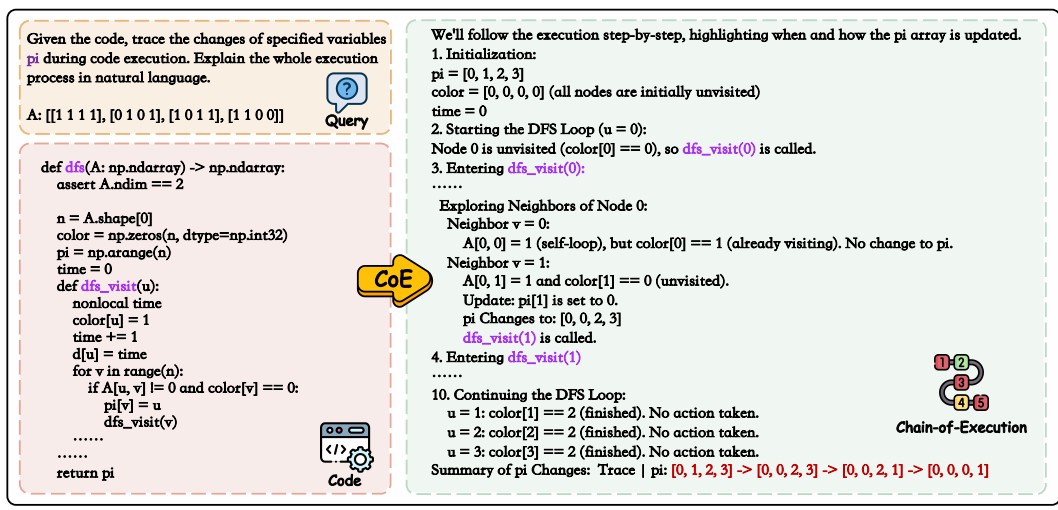

Figure 2: A classical DFS algorithm example of CoE in TracePile. More cases are in Appendix C.

**Query Diversification.** To move beyond the standard function-level input/output prediction tasks, we leverage Qwen-2.5-72B-Instruct to generate alternative queries for a given code snippet. Traditional algorithmic queries typically ask for the function's return value. To increase reasoning granularity, we prompt the model to generate questions targeting internal execution states. For instance, in a DFS implementation, the model might be asked to predict the stack contents after a specific backtracking step or trace the change in a visited-node array. These questions compel the model to understand intermediate logic and control flow, promoting deeper procedural reasoning. We apply this strategy to algorithmic and graph problems but not to mathematical code, which often has a single well-defined objective and is less amenable to meaningful query variation.

**Code Diversification.** To increase structural variety, we also instruct Qwen-2.5-72B-Instruct to rewrite the original code implementations. For example, a recursive DFS might be rephrased into an iterative version using an explicit stack, or a dynamic programming solution might be restructured to vary loop orders or base case initialization. Such rewrites preserve semantics while introducing syntactic and structural variation, helping the model generalize beyond rigid code templates and better handle real-world coding diversity. See related prompts in Appendix A.

While rejection sampling is a common method for increasing solution diversity, we choose not to apply it in the main TracePile construction for two reasons. First, our query and code augmentations already induce diverse CoE samples implicitly. Second, given the wide coverage of mathematical topics in our dataset, CoE formats significantly, making it difficult to apply rejection filtering in a scalable and reliable manner. Nonetheless, we explore rejection sampling as a potential enhancement in later experiments and discuss its effects in Section 4.

## 2.3 Chain of Execution Generation

Once we obtain the curated query-code pairs, we generate detailed CoE data using Qwen-2.5-72B-Instruct under carefully designed prompting strategies. Our goal is to convert these execution behaviors into coherent, step-by-step explanations in natural language that reflect fine-grained procedural reasoning.

The prompting format is tailored slightly based on the data source and task type, but generally follows patterns such as: *"Answer the question by tracing the code execution step by step." "Trace the changes of variable X during execution and explain each step."*

To ensure the generated CoE outputs are logically consistent and sufficiently detailed, we employ two key strategies: (1) Few-shot Prompting. We provide two human-written in-context examples within each prompt, showcasing complete step-by-step CoE narrations for similar tasks. These exemplars guide the model in understanding both the structure and the level of granularity expected in the reasoning process. (2) Intermediate result verification. For tasks requiring variable tracking or control-flow interpretation, we require the model to output structured *json* results. For mathematical problems,

Table 1: Model performances under continue-pretraining. L.C.Bench-O refers to the test-output subset of LiveCodeBench. The best results are in purple.

| Type | Datasets | Llama-3-8B | | Llama-3.1-8B | | Qwen-2.5-7B | | Qwen-2.5-Coder | |
|---|---|---|---|---|---|---|---|---|---|
| | | Base | Ours | Base | Ours | Base | Ours | Base | Ours |
| Math | GSM8K | 54.2 | **74.3** | 54.4 | **74.7** | **85.7** | 84.9 | 77.9 | **83.5** |
| | MATH | 16.5 | **31.7** | 17.7 | **29.3** | **50.9** | 44.8 | 47.2 | **52.3** |
| | GSM-H | 26.1 | **36.2** | 27.1 | **36.9** | **63.3** | 63.1 | 55.1 | **59.8** |
| | SVAMP | 68.8 | **81.2** | 71.0 | **81.1** | 89.4 | **89.6** | 87.8 | **89.0** |
| | ASDIV | 73.1 | **82.7** | 74.3 | **83.8** | 91.0 | **91.3** | 89.0 | **89.2** |
| | MAWPS | 90.9 | **92.2** | 92.0 | **93.7** | 97.0 | **97.1** | 93.5 | **94.9** |
| | STEM | 49.7 | **56.3** | 57.0 | **57.1** | 68.0 | **72.6** | 67.2 | **68.3** |
| | TABMWP | **57.9** | 55.0 | **63.6** | 57.4 | **73.0** | 72.3 | 56.9 | **66.2** |
| | SAT | 56.2 | **59.4** | 59.4 | **68.8** | 80.0 | **93.8** | 81.2 | **84.4** |
| | Average | 54.8 | **63.4** | 57.4 | **64.8** | 77.6 | **78.8** | 72.9 | **76.4** |
| Code | L.C.Bench-O | 2.0 | **30.5** | 1.6 | **11.8** | 40.3 | **49.8** | 14.0 | **44.8** |
| Logical | Zebra Puzzle | 0.1 | **7.4** | 2.3 | **7.4** | **2.9** | 2.0 | 3.0 | **4.0** |
| | KORBench | **21.8** | **21.8** | 21.9 | 20.6 | 33.4 | **35.4** | **32.0** | 30.0 |
| | Ruletaker | 2.8 | **42.4** | 5.3 | **45.6** | 61.2 | **63.5** | 46.1 | **60.2** |
| Algorithm | Graphwiz | 4.5 | **46.0** | 1.9 | **33.9** | 36.6 | **50.5** | 38.5 | **43.8** |
| | GraphInstruct | 35.2 | **73.5** | 33.0 | **69.2** | 33.1 | **33.5** | 33.8 | **53.7** |

the final computed result is reported in json to simplify correctness checking. For algorithmic tasks, the json includes both the final output and intermediate variable states (e.g., array contents, stack traces) across execution steps—formatted as "$state_1$ -> $state_2$ -> ...$state_n$"—to facilitate automatic comparison with execution logs. We validate correctness by comparing these outputs to ground-truth traces generated via code instrumentation or symbolic tools, discarding samples that fail consistency checks. We showcase an example of CoE in Figure 2. In the final version of our dataset, we convert the json outputs back into natural language format. This adjustment was made after observing that retaining structured json caused the model to produce inconsistent or malformed outputs on some downstream datasets after continued pretraining.

To improve coverage on harder problems, we sample each CoE rationale five times independently. Some tasks are too challenging for the model to solve correctly in a single attempt, so multiple samples increase the chance of getting a valid solution.

Finally, we control for CoE complexity. As input size increases—particularly in graph algorithms—the reasoning path and solution length often grow rapidly. For example, denser or larger graphs lead to execution traces that scale combinatorially, often resulting in CoE outputs exceeding tens of thousands of tokens. Since such long outputs are prone to generation errors and misalignment, we impose a strict 8k-token limit, consistent with the context length used during continue-pretraining. Any sample exceeding this limit is discarded to maintain training efficiency and consistency.

## 2.4 Statistics

We summarize the composition of TracePile in Table 2, which includes three primary components: algorithmic competition data, classical algorithm implementations, and mathematical code. In total, TracePile contains over **2.6** million CoE samples comprising approximately **19** billion tokens. The algorithmic competition subset contributes 480K samples sourced from Codeforces, covering diverse real-world

Table 2: Statistics of TracePile.

| Components | Size | Tokens |
|---|---|---|
| Algorithmic Competition | 480,782 | 3,891,991,488 |
| Classical Algorithm | 949,088 | 7,024,210,656 |
| Mathematical | 1,170,836 | 7,105,488,512 |
| **Total** | **2,600,706** | **19,380,461,000** |

problem-solving code. The classical algorithm subset includes 949K structured samples based on 30 canonical algorithms. The mathematical subset consists of 1.17M samples drawn from OpenMath, emphasizing multi-step symbolic computation. Notably, the classical algorithm and

Table 3: Model performances under two-stage fine-tuning. The base models are trained solely on TuluSFT datasets. L.C.Bench refers to LiveCodeBench. The best results are in purple.

| Type | Datasets | Llama-3-8B | | Llama-3.1-8B | | Qwen-2.5-7B | | Qwen-2.5-Coder | |
| --- | --- | --- | --- | --- | --- | --- | --- | --- | --- |
| | | Base | Ours | Base | Ours | Base | Ours | Base | Ours |
| Math | GSM8K | 70.7 | **74.0** | 73.8 | **76.1** | 76.9 | **77.0** | 74.3 | **80.8** |
| | MinMath | 28.8 | **33.8** | 30.2 | **33.2** | 47.2 | **51.8** | **47.6** | 48.2 |
| | MATH | 28.6 | **30.6** | 30.8 | **34.2** | 49.6 | **51.5** | 45.0 | **50.8** |
| | GSM-H | 35.0 | **35.6** | 39.3 | 35.3 | 55.0 | **55.6** | 55.4 | **59.8** |
| | SVAMP | **83.0** | 79.4 | 79.6 | **83.4** | 79.7 | **83.6** | 81.1 | **85.3** |
| | ASDIV | 81.4 | **81.9** | 82.3 | **85.7** | **86.5** | 86.3 | 86.9 | **88.9** |
| | MAWPS | **93.9** | 93.5 | **94.3** | 93.1 | 95.6 | **95.9** | 95.4 | **96.6** |
| | STEM | 42.8 | **52.9** | 50.2 | **58.9** | 70.1 | **71.9** | 69.3 | 66.2 |
| | TABMWP | 65.8 | **67.1** | **69.6** | 57.2 | 77.3 | **78.9** | 80.5 | 78.3 |
| | MATHQA | 37.7 | **54.7** | 44.7 | **60.2** | **80.7** | 80.2 | 75.3 | **77.2** |
| | SAT | 40.6 | **75.0** | 65.6 | **68.8** | 90.6 | **96.9** | 87.5 | 84.4 |
| | Average | 55.3 | **61.7** | 60.0 | **62.4** | 73.6 | **75.4** | 72.6 | **74.2** |
| Code | CRUX | 32.9 | **42.1** | 33.5 | **49.9** | 46.5 | **49.4** | 52.8 | **56.4** |
| | L.C.Bench | 9.8 | **14.2** | 7.6 | **11.3** | **25.8** | 23.5 | 27.2 | **29.9** |
| Logical | Zebra Logic | 7.7 | **10.5** | **8.1** | 9.4 | **9.4** | 8.8 | 9.2 | **10.8** |
| | KORBench | **27.0** | 27.2 | **27.0** | 24.7 | 40.4 | **41.0** | 34.6 | **39.4** |
| | Mmlu-Redux | 51.8 | **56.3** | 53.8 | **54.3** | 64.2 | **69.7** | 66.5 | 64.2 |
| | Ruletaker | 60.4 | **66.2** | 61.5 | **67.0** | 73.2 | **74.4** | 73.1 | **73.9** |
| Algorithm | Graphwiz | **44.0** | 39.6 | **44.5** | 39.2 | 29.0 | **33.2** | 33.9 | **35.4** |
| | GraphInstruct | 27.5 | **36.0** | 29.8 | **55.9** | 32.4 | **35.8** | 31.5 | **40.9** |
| | CLRS | 17.2 | **24.5** | 16.8 | **25.8** | 36.0 | **43.6** | 42.2 | **49.4** |

algorithmic competition portion contribute the largest share of tokens, reflecting the high density of intermediate reasoning in those samples. This large and diverse corpus forms the foundation for studying general-purpose reasoning enhancement through code execution supervision.

## 3 Experiments

In this section, we first introduce the experimental settings, including training setup, and evaluation datasets. Then, we evaluate the effectiveness of TracePile in 20 datasets under three training setups.

### 3.1 Experimental Settings

**Training Setups.** Our goal is to investigate whether fine-grained, step-by-step execution data can serve as an effective intermediate training signal to enhance multi-domain reasoning performance. To this end, we design three complementary training settings: (1) **Continue-pretraining on TracePile**, which introduces structured reasoning patterns into the base model and strengthens its ability to perform logical, step-by-step inference. (2) **Instruction tuning after pretraining**, which ensures that the model retains broad instruction-following capabilities while benefiting from the reasoning skills acquired during TracePile continue-pretraining checkpoints. (3) **Two-stage instruction-tuning**, where the model is first trained on TracePile and then further tuned on general instruction datasets. This staged adaptation enables the integration of specialized reasoning without compromising general alignment, resulting in a more balanced and capable model across tasks.

To ensure the generality and robustness of our findings, we select four representative **base models** for evaluation: LLaMA 3-8B, LLaMA 3.1-8B, Qwen-2.5-7B, and Qwen-2.5 Coder-7B. These models span both general-purpose and code-oriented architectures, allowing us to assess the effectiveness of TracePile across diverse model families and training paradigms. For all training experiments—including both continue-pretraining and instruction tuning—we use 16 H800-80GB GPUs with a batch size of 512, a maximum sequence length of 8192 tokens, and 3 training epochs. The learning rate is set to 1e-5, and all training is conducted using the LLaMA-Factory framework. For the general instruction tuning phase, we adopt the widely used Tulu3-SFT [26] dataset to ensure

broad instruction coverage and compatibility with existing evaluation standards. This consistent and scalable setup enables controlled comparisons across models and training strategies.

**Evalution Dataset.** To comprehensively assess the reasoning capabilities of TracePile-enhanced models and strong baselines, we evaluate across 20 benchmarks spanning four major reasoning domains: **logical reasoning**, **algorithmic reasoning**, **code reasoning**. For mathematical reasoning, we select 11 diverse datasets: GSM8K [8], MATH [20], GSM8K-Hard [14], SVAMP [36], ASDIV [35], MAWPS [25], MinMATH [20], MMLU-STEM [19], TABMWP [30], MATHQA [1], and SAT [55]. For logical reasoning, we include Zebra Logic [29], RuleTaker [6], and KORBench [33]. For code reasoning, we evaluate on LiveCodeBench [22] and CRUX [16] . For algorithmic reasoning, we adopt GraphWiz [3], GraphInstruct [32], and CLRS [34]. We aggregate these datasets through three public evaluation toolkits: OpenCompass[9], Qwen2.5-Math [50], and ZeroEval [28], ensuring consistency and reproducibility across experiments. **See Appendix B for details of evaluation datasets.**

*It is worth noting that we observe poor instruction-following behavior in some base models that have not undergone instruction tuning*. For datasets with complex or structured instructions—such as CLRS—we exclude evaluation under the continue-pretraining-only setting, as the models are unable to generate meaningful outputs without prior alignment.

## 3.2 Main Results

The experimental results in Tables 1 and 3 demonstrate the effectiveness of TracePile across both continue-pretraining and two-stage instruction-tuning settings. Under continue-pretraining, models consistently gain significant improvements across mathematical, logical, algorithmic, and code reasoning tasks. For example, LLaMA-3-8B improves by +8.4% on average over nine math benchmarks (from 54.8% to 63.4%), with particularly large gains on GSM8K. Similarly, Qwen-2.5-Coder shows strong improvements across domains, indicating that even code-specialized models benefit from fine-grained execution supervision. TracePile proves especially effective on algorithmic reasoning, with improvements exceeding +40% on tasks such as GraphWiz and GraphInstruct. When combined with general instruction tuning in the two-stage finetuning setting, these improvements persist and extend further. For instance, LLaMA-3-8B achieves a +6.4% average gain (from 55.3% to 61.7%) on math datasets, and Qwen-2.5-7B improves from 73.6% to 75.4%. **Due to space limitations, we report results of continue-pretraining + instruction tuning setting in the Appendix, Table 7**, which further corroborate the value of TracePile as a general-purpose reasoning enhancement stage.

**Out-of-domain (OOD) Generalization.** Notably, TracePile also improves performance on benchmarks with little direct data or no overlap, such as LiveCodeBench, CRUX, and logical reasoning datasets like RuleTaker and Zebra Logic. The resulting models' performances continue to improve on these datasets. These tasks differ in format, domain, or reasoning style from TracePile's training sources, yet models still show consistent improvements. This suggests that the benefits of explicit, execution-based supervision generalize beyond the original domains, helping models develop transferable reasoning skills. The ability to improve both in-domain and out-of-domain tasks highlights TracePile's effectiveness as a general-purpose reasoning enhancement corpus.

## 4 Discussion

To further understand the effectiveness and generalization behavior of TracePile, we conduct a series of analysis experiments. These analyses aim to evaluate the contribution of key design choices (e.g., step-by-step supervision, data diversity), the scalability of TracePile, its robustness, and its transferability beyond code-related tasks. Below, we address several core questions that deepen our understanding of TracePile's impact.

**RQ1: How important is the Chain of Execution (CoE) format compared to traditional I/O or solution-style supervision?** To evaluate the contribution of our step-by-step CoE supervision, we compare TracePile with several strong baselines based on Qwen-2.5-Coder under a controlled two-stage finetuning setting. As shown in Table 4, we include three relevant baselines: (1) OpenMathInstruct-1, where we replace our CoE-based math data with the original solution-style code from OpenMath; (2) OpenMathInstruct-2, where we replace our CoE-based math data with the pure math CoT-style solutions; (3) Webinstruct, where we replace our CoE-based math data with the generic instruction data from WebInsrtuct; CodeI/O [27], which contains 3.5M input-output code

samples with final-answer supervision; and (4) CodeI/O++, a lightly augmented variant with minimal structural enhancements.

Despite being smaller or equal in size, TracePile outperforms all baselines across both in-domain and out-of-domain reasoning benchmarks. On GSM8K, TracePile achieves 80.8%, slightly above OpenMath (80.1%) and CodeI/O++ (80.5%). On MATH, which demands more compositional reasoning,

Table 4: Comparison of TracePile with alternative supervision strategies under two-stage fine-tuning on Qwen-2.5-Coder 7B. LC-O and Z.L denote LiveCodeBench-Output and Zebra Logic.

| Methods | Stage-1 Data | GSM8K | MATH | LC-O | Z.L |
|---|---|---|---|---|---|
| Baseline | – | 74.3 | 45.0 | 29.8 | 9.2 |
| Pure Code Solution | OpenMathIns. 1 | 80.1 | 46.1 | 19.4 | 5.6 |
| Pure Math Solution | OpenMathIns. 2 | **82.1** | **51.1** | 17.4 | 6.6 |
| Generic Instruction | WebInstruct | 81.3 | 49.0 | 18.8 | 9.1 |
| CodeI/O | CodeI/O | 79.3 | 39.8 | 17.6 | 8.6 |
| **Ours** | TracePile | 80.8 | 50.8 | **35.5** | **10.8** |

the improvement is more substantial—50.8% for TracePile versus 46.1% (OpenMath) and only 39.9% (CodeI/O++). The advantage of CoE becomes even more pronounced on out-of-domain tasks. For example, on LiveCodeBench-Output, TracePile reaches 35.5%, far surpassing CodeI/O++ (20.6%) and OpenMath (3.4%). On Zebra Logic, TracePile scores 10.8%, outperforming all baselines. These results confirm that the Chain of Execution format—by explicitly guiding models through intermediate reasoning steps—offers more effective supervision than final-answer training. TracePile's CoE data not only enhances in-domain learning (e.g., math, algorithms) but also enables strong generalization to unseen task formats, making it a powerful training resource for robust and transferable reasoning.

**RQ2: What components contribute most to TracePile's effectiveness?**  To assess the contribution of different components within TracePile, we conduct ablation studies on LLama-3.1-8B under the two-stage finetuning setting. Specifically, we remove individual data sources or design features and evaluate their impact on downstream performance across four reasoning domains: mathematical, code, logical, and algorithmic. The results are summarized in Table 5.

The ablation results clearly demonstrate that both the multi-source composition and the diversification strategies within TracePile are crucial to its effectiveness as an intermediate reasoning stage. Removing any individual data source—mathematical, algorithmic, or competition code—leads to noticeable performance degradation, not only within the corresponding domain but also across others, highlighting strong

Table 5: Ablation studies under two-stage fine-tuning. Average performances of each category are reported.

| Methods | Mathematical | Code | Logical | Algorithm |
|---|---|---|---|---|
| **Ours** | **62.4** | **30.6** | **38.9** | 40.3 |
| w/o mathematical | 60.6 | 26.2 | 38.4 | **40.6** |
| w/o algorithm | 61.5 | 23.1 | 38.0 | 33.2 |
| w/o competition | 62.0 | 24.9 | 37.8 | 35.0 |
| w/o diversification | 61.8 | 27.6 | 38.0 | 37.2 |
| w/o query | 62.0 | 30.0 | 38.7 | 37.9 |
| w/o code | 62.2 | 28.2 | 38.3 | 39.5 |

cross-domain transfer. For example, excluding competition data significantly weakens code reasoning (−5.7%), emphasizing its role in grounding execution-based understanding. Diversification strategies further enhance generalization: eliminating both query and code augmentations reduces performance across all categories, particularly in code (−3.0%) and algorithmic (−3.1%) reasoning. Among them, code rewrites contribute most to structural generalization, while query diversification enhances task-specific interpretability. These findings reinforce that TracePile's effectiveness is not solely due to data scale, but arises from its carefully constructed CoE supervision, domain diversity, and reasoning-aware augmentation.

**RQ3: Does performance scale with more TracePile data?**  To examine how performance scales with data size, we progressively expand TracePile from 50K to 4.3M samples and evaluate model performance using LLaMA-3.1-8B under the two-stage finetuning setup. As shown in Figure 3, we

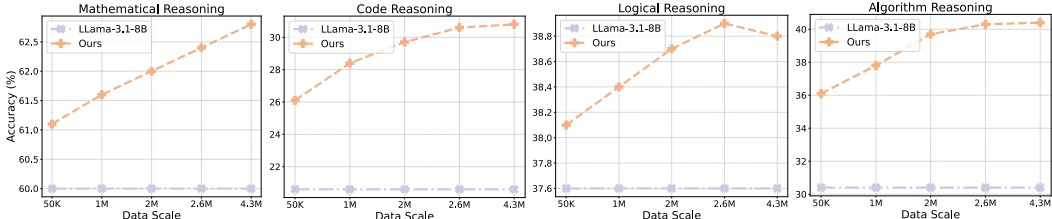

Figure 3: Model performances with scaling the size of CoE samples.

observe a clear upward trend across all four reasoning domains: mathematical, code, logical, and algorithmic. This confirms that increasing the volume of CoE supervision directly contributes to stronger reasoning capabilities, especially in domains with structured multi-step logic.

To further explore the effect of high-quality data at scale, we apply rejection sampling to augment TracePile to 4.3M samples, denoted as **TracePile++**. The goal is to filter for higher-quality CoE traces by selecting the diverse outputs from multiple samples. Interestingly, while mathematical reasoning continues to benefit from this expansion, the improvements in code, logical, and algorithmic reasoning plateau or slightly regress. This suggests that rejection sampling, despite improving local coherence, may introduce distributional biases that limit broader generalization.

**RQ4: What types of reasoning are improved by TracePile? Does it enhance robustness, reduce typical reasoning errors, and generalize?** To better understand the qualitative improvements introduced by TracePile, we conduct a fine-grained analysis on several challenging reasoning subsets from the BBH benchmark [41]. These tasks are carefully chosen to test specific dimensions of reasoning where CoE-style supervision may offer unique advantages. We focus on the following four subsets: *Tracking Shuffled Objects*: Requires precise state tracking across sequential transformations (e.g., object swaps), a skill closely aligned with the procedural traceability in TracePile. *Multi-Step Arithmetic*: Involves multi-hop numerical reasoning across chained operations. *Logical Deduction*: Demands inference from spatial or relational constraints to determine object orderings—akin to graph- or rule-based reasoning patterns. *Web of Lies*: Evaluates truth values from complex Boolean logic expressed in natural language, testing abstract symbolic manipulation.

As shown in Table 6, models trained on TracePile outperform the Qwen-2.5-base model across all tasks. The gains are particularly striking in Tracking Shuffled Objects (+19.2%) and Web of Lies (+5.6%), suggesting improved robustness in stateful and symbolic reasoning. TracePile also boosts performance in Multi-

Table 6: Compare ours with Qwen-2.5-base under two-stage fine-tuning in BBH subsets.

| **Modelsm** | Tracking Shuffled Objects | Arithmetic | Logical Deduction | Web of Lies |
|---|---|---|---|---|
| Base | 68.4 | 72.0 | 78.4 | 87.2 |
| **Ours** | **87.6** | **82.0** | **83.6** | **92.8** |

Step Arithmetic (+11.6%) and Logical Deduction (+3.6%), where intermediate reasoning steps are critical. These results provide evidence that TracePile significantly enhances fine-grained reasoning abilities. It helps models better track latent state, execute multi-step logic, and reduce common failure cases like missing intermediate computations or making logically inconsistent predictions. More importantly, it shows that CoE-based supervision transfers beyond code tasks, improving general reasoning robustness across abstract formats.

## 5 Related Works

**Reasoning in Large Language Models.** Reasoning is a core capability for large language models (LLMs) and a critical step toward achieving general intelligence. Recent work has shown that LLMs possess some capacity for complex reasoning when prompted appropriately, leading to a surge of interest in improving their reasoning abilities, particularly in domains such as mathematical problem solving and code generation [48, 37, 4, 5, 1, 21]. Early efforts often relied on prompt engineering techniques, such as chain-of-thought prompting [49], self-consistency, and tool-augmented scratchpad

[46], to elicit multi-step reasoning behavior at inference time. More recently, the field has shifted toward data-driven approaches, where large-scale, domain-specific reasoning corpora are collected and used for pretraining or finetuning. Representative examples include Qwen-Math [51], Qwen-Coder [18], DeepSeek-Math [39], and DeepSeek-Coder [21], which leverage hundreds of billions to trillions of tokens to specialize models for mathematical or programming tasks. Compared to these works, TracePile introduces a different perspective: rather than relying solely on final-answer or solution-style supervision, it provides fine-grained, step-by-step execution traces in natural language through the CoE format. This enables more explicit and interpretable reasoning supervision, bridging the gap between task-specific pretraining and general-purpose reasoning enhancement.

**Supervision through Code Execution.** The idea of learning from code execution traces predates the modern era of LLMs, with early work exploring how neural networks could simulate or reason over program behavior [23, 54]. More recently, this line of research has been revived in the context of LLMs, but most existing efforts focus narrowly on output prediction from code execution [13]. Other approaches aim to leverage execution in auxiliary ways—either as final feedback for reward learning [11] or by incorporating intermediate traces to improve code generation [12]. SemCoder [12] employs "monologue reasoning" with high-level functional descriptions and verbal reasoning about local execution effects. However, it primarily uses synthetic data with limited real-world complexity. Additionally, new benchmarks such as CRUXEval [16] have been introduced to test models' ability to simulate or predict execution dynamics. Most recently, CodeI/O [27] also proposes code-referenced input-output pairs to enhance the models' reasoning abilities. In contrast to these code-centric and task-specific efforts, TracePile takes a broader view. It is the first to train LLMs on large-scale, diverse execution traces using natural language CoE supervision.

# 6 Conclusion

In this work, we present TracePile, a large-scale dataset of step-by-step Chain of Execution (CoE) traces designed to enhance the reasoning abilities of large language models, covering over 2.6 million samples. By supervising models with fine-grained execution processes across mathematics, classical algorithms, and competition code, TracePile provides a structured reasoning signal that extends beyond final-answer supervision. Our experiments across multiple training strategies and 20 diverse benchmarks demonstrate that TracePile consistently improves both in-domain and out-of-domain reasoning, especially on tasks requiring multi-step logic and state tracking. Ablation studies further highlight the importance of multi-source design and diversification strategies, while scaling analysis confirms that performance improves with data volume, though with diminishing returns beyond a certain point. Overall, TracePile offers an effective and transferable intermediate training stage that significantly boosts the general reasoning capabilities of LLMs.

# Limitation

While TracePile demonstrates strong improvements in general reasoning, it also comes with certain limitations. First, the dataset is primarily grounded in code-based reasoning, which may limit its applicability to tasks requiring commonsense or world knowledge that fall outside structured procedural logic. Second, although we incorporate multiple domains, the coverage is still skewed toward mathematical and algorithmic reasoning; expanding to domains such as law, planning, or real-world causal inference remains future work. Third, generating high-quality CoE traces relies heavily on large instruction-tuned models, which introduces computational costs and potential bias from the prompting model itself. Lastly, our filtering strategy (e.g., 8k token cutoff) may discard complex but informative reasoning traces, potentially limiting exposure to long-range dependencies. Addressing these challenges—such as improving coverage, reducing reliance on frontier models, and capturing longer CoE traces—will be critical for further scaling TracePile's effectiveness.

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

# A Prompts

---

**Prompts A.1: Prompt of LLM extracting clean code**

You are an AI assistant responsible for providing assistance. You are provided with a problem and examples of the input.
Your task is: (1) Write Python code to develop an input generator function capable of generating inputs for this problem. The input generator function should include a parameter that determines the maximum number of inputs it generates. The output of this function should be a list containing the generated inputs, with the length of each individual input not exceeding 10. Besides, any numbers contained in the input should not exceed 1000. Please provide the Python code without any additional explanation. Present your output in the following format: Input_generator: <your code>
(2) Rewrite the code to eliminate manual input and encapsulate it within a function named 'solution'. The function should accept a string input as its parameter. Provide the output in the following format: Solution: <your code>.

Here is relative information:
Original Problem: {Original Problem}
Input: {Input examples}

---

**Prompts A.2: An Example of LLM Rewrite Prompt**

You are provided with Python code that solves a specific problem. Your task is to rewrite the code while adhering to the following guidelines:
(1) Retain the original logic and functionality of the provided code.

---

(2) Introduce variations in the implementation, such as using different syntax, alternative methods, or restructuring the code for improved readability or efficiency.
(3) Ensure the rewritten code is clean, functional, and adheres to best practices.

Output only the modified code. Do not include any explanations or comments.

Here is relative information:
Code: {Code}

## B Benchmark Overview

To comprehensively evaluate the reasoning capabilities of our models, we select **20 benchmark datasets** spanning four major reasoning domains: *mathematical reasoning*, *logical reasoning*, *algorithmic and code reasoning* Below, we briefly describe each dataset:

**GSM8K**  A collection of 1K grade school math word problems requiring multi-step arithmetic reasoning. Designed to evaluate basic numerical reasoning skills.

**MATH**  A large-scale dataset of 12.5K competition-style math problems covering algebra, geometry, number theory, and combinatorics. It tests advanced symbolic and procedural reasoning.

**GSM8K-Hard**  A harder version of GSM8K where numbers are replaced with uncommon or large values, increasing arithmetic and symbolic complexity.

**SVAMP**  An elementary-level math benchmark that tests a model's sensitivity to problem structure and its ability to reason over similar surface forms with different logical requirements.

**ASDIV**  A diverse dataset of 2.3K elementary math word problems drawn from textbooks, focusing on linguistic variation and real-world phrasing.

**MAWPS**  A corpus of 3.3K math problems scraped from educational resources, covering various question types used in real-world math education.

**MINERVA_Math**  A curated set of 272 high-complexity math questions, emphasizing multi-step symbolic manipulation and abstract reasoning.

**MMLU-STEM**  A subset of the MMLU benchmark covering science, technology, engineering, and mathematics subjects. Includes questions from high school and university-level curricula.

**TABMWP**  A structured math word problem dataset (8.5K samples) formulated in tabular format, designed to test abstract reasoning and table comprehension.

**MATHQA**  A dataset of 3K math word problems annotated with symbolic program representations, enabling reasoning over structured solution steps.

**SAT-MATH**  A dataset mimicking standardized SAT math questions, covering algebra, geometry, and data analysis. Designed to test broad quantitative proficiency.

**Zebra Puzzle**  A logic puzzle dataset derived from constraint satisfaction problems. Tasks require deducing object relationships and properties based on textual clues.

**Ruletaker**  A benchmark that presents logical rules and facts in natural language and asks models to infer new truths through deductive reasoning.

**ProofWriter**  Includes small rulebases of English facts and logical rules, where models must determine whether a hypothesis is provable, unprovable, or unknown.

**CLRS**  Based on the "Introduction to Algorithms" textbook, CLRS provides algorithmic tasks where models are expected to simulate or reason about classical algorithm behavior.

**GraphWiz**  Contains 3.6K problems across nine graph reasoning tasks (e.g., shortest path, reachability), with complexity ranging from linear to NP-complete.

**GraphInstruct**  A synthetic benchmark consisting of 21 graph-based reasoning tasks. We evaluate on a subset aligned with general reasoning patterns.

**KorBench**  KorBench [33] is designed to assess a model's intrinsic reasoning and planning abilities while minimizing the influence of pretrained world knowledge. It introduces five categories—*Operation*, *Logic*, *Cipher*, *Puzzle*, and *Counterfactual*—each defined by 25 manually constructed, novel rules. This setup enables a more precise evaluation of a model's ability to generalize to unfamiliar, rule-based tasks.

**CRUXEval**  CRUXEval evaluates a model's capability to reason over code by predicting either inputs or outputs of anonymized Python functions. This benchmark requires models to perform symbolic execution and maintain intermediate state across program flow.

**BBH**  BBH (Beyond the Imitation Game Benchmark) comprises 23 challenging reasoning tasks originally from the BIG-Bench suite. These tasks are curated to be particularly difficult for LLMs, covering a wide range of logical, mathematical, and procedural reasoning problems.

Together, these benchmarks provide a comprehensive and diverse testbed for evaluating general reasoning across symbolic, procedural, logical, and open-domain tasks.

Table 7: Model performances under contiue pretraining+fine-tuning settings. The base models are trained on the same dataset. L.C.Bench refers to livecodebench.

| Type | Datasets | Llama-3-8B | | Llama-3.1-8B | | Qwen-2.5-7B | | Qwen-2.5-Coder | |
|---|---|---|---|---|---|---|---|---|---|
| | | Base | Ours | Base | Ours | Base | Ours | Base | Ours |
| Math | GSM8K | 70.7 | **72.0** | 73.8 | **78.9** | **76.9** | 76.9 | 74.3 | **80.8** |
| | MinMath | 28.8 | **29.8** | **30.2** | 30.0 | 47.2 | **52.8** | 47.6 | **50.4** |
| | MATH | 28.6 | **29.1** | 30.8 | **31.9** | 49.6 | **51.6** | 45 | **50.2** |
| | GSM-H | **35.0** | 34.0 | **39.3** | 38.2 | **55.0** | 54.1 | 55.4 | **59.8** |
| | SVAMP | **83.0** | 79.9 | 79.6 | **83.4** | 79.7 | **81.4** | 81.1 | **83.0** |
| | ASDIV | 81.4 | **82.2** | 82.3 | **84.7** | 86.5 | **88.3** | 86.9 | **88.2** |
| | MAWPS | 93.9 | **94.1** | **94.3** | 94.2 | 95.6 | **95.7** | 95.4 | **95.8** |
| | STEM | 42.8 | **52.8** | 50.2 | **58.8** | 70.1 | **71.4** | **69.3** | 67.3 |
| | TABMWP | 65.8 | **68.0** | 69.6 | **72.0** | **77.3** | 76.5 | **80.5** | 79.5 |
| | MATHQA | 37.7 | **49.6** | 44.7 | **59.8** | **80.7** | 80.6 | 75.3 | **78.0** |
| | SAT | 40.6 | **62.5** | **65.6** | 65.6 | **90.6** | 87.5 | 87.5 | **90.6** |
| | Average | 55.3 | **59.5** | 60.0 | **63.4** | 73.6 | **74.3** | 72.6 | **74.9** |
| Code | CRUX | 32.9 | **34.4** | 34.0 | **40** | 46.5 | **49.8** | 52.6 | **58.1** |
| | L.C.Bench | 9.8 | **10.1** | 7.6 | **14.7** | **25.8** | 23.9 | 27.2 | **29.0** |
| Logical | Zebra Puzzle | 7.7 | **8.9** | 8.1 | **10.7** | **9.4** | 8.9 | 9.2 | **9.6** |
| | KORBench | 47.6 | **75.6** | **27.0** | 24.7 | 40.4 | **41.5** | 34.6 | **35.9** |
| | Ruletaker | 60.4 | **65.4** | 61.5 | **65.1** | 73.2 | **74.7** | **73.1** | 72.4 |
| Algorithm | Graphwiz | **44.0** | 29.8 | **44.5** | 43.1 | 29.0 | **70.9** | 33.9 | **61.9** |
| | GraphInstruct | 27.5 | **44.0** | 29.8 | **55.0** | 32.4 | **63.6** | 31.5 | **62.2** |
| | CLRS | 17.2 | **32.5** | 16.8 | **38.3** | 36.0 | **52.0** | 41.2 | **58.9** |

## C  Cases in TracePile

---

**Prompts C.1: An Case from Classical algorithm in TracePile**

Given the code, trace the changes of specified variables during code execution.

Input:
```python
def dfs(A: np.ndarray) -> np.ndarray:
    assert A.ndim == 2

    n = A.shape[0]
    color = np.zeros(n, dtype=np.int32)
    pi = np.arange(n)
    d = np.zeros(n)
    f = np.zeros(n)
    time = 0

    def dfs_visit(u):
        nonlocal time
        color[u] = 1
        time += 1
        d[u] = time

        for v in range(n):
            if A[u, v] != 0 and color[v] == 0:
                pi[v] = u
                dfs_visit(v)

        color[u] = 2
        time += 1
        f[u] = time

    for u in range(n):
        if color[u] == 0:
            dfs_visit(u)

    return pi
```
along with the input parameter: A: [[1 1 1 1], [0 1 0 1], [1 0 1 1], [1 1 0 0]] Trace the changes in the "pi" variable throughout the program's execution.
**CoE**:

We'll follow the execution step-by-step, highlighting when and how the pi array is updated.
```
1. Initialization:

pi = [0, 1, 2, 3]
color = [0, 0, 0, 0] (all nodes are initially unvisited)
time = 0

2. Starting the DFS Loop (u = 0):

Node 0 is unvisited (color[0] == 0), so dfs_visit(0) is called.

3. Entering dfs_visit(0):
    color[0] is set to 1 (visiting).
    time increments to 1.
    d[0] = 1.

  Exploring Neighbors of Node 0:
    Neighbor v = 0:
        A[0, 0] = 1 (self-loop), but color[0] == 1 (already visiting
            ). No change to pi.
    Neighbor v = 1:
```

---

```
        A[0, 1] = 1 and color[1] == 0 (unvisited).
        Update: pi[1] is set to 0.
        pi Changes to: [0, 0, 2, 3]
        dfs_visit(1) is called.

4. Entering dfs_visit(1):

   color[1] is set to 1 (visiting).
   time increments to 2.
   d[1] = 2.
 Exploring Neighbors of Node 1:

   Neighbor v = 0:
       A[1, 0] = 0 (no edge). No change to pi.
   Neighbor v = 1:
       A[1, 1] = 1 (self-loop), but color[1] == 1 (already visiting
           ). No change to pi.
   Neighbor v = 2:
       A[1, 2] = 0 (no edge). No change to pi.
   Neighbor v = 3:
       A[1, 3] = 1 and color[3] == 0 (unvisited).
       Update: pi[3] is set to 1.
       pi Changes to: [0, 0, 2, 1]
       dfs_visit(3) is called.
5. Entering dfs_visit(3):

   color[3] is set to 1 (visiting).
   time increments to 3.
   d[3] = 3.
 Exploring Neighbors of Node 3:

   Neighbor v = 0:
       A[3, 0] = 1 and color[0] == 1 (already visiting). No change
           to pi.
   Neighbor v = 1:
       A[3, 1] = 1 and color[1] == 1 (already visiting). No change
           to pi.
   Neighbor v = 2:
       A[3, 2] = 0 (no edge). No change to pi.
   Neighbor v = 3:
       A[3, 3] = 0 (no edge). No change to pi.
       Post Exploration:
       color[3] is set to 2 (finished).
       time increments to 4.
       f[3] = 4.
       pi Remains: [0, 0, 2, 1]

6. Returning to dfs_visit(1):

   Post Exploration:
       color[1] is set to 2 (finished).
       time increments to 5.
       f[1] = 5.
       pi Remains: [0, 0, 2, 1]

7. Returning to dfs_visit(0):

   Neighbor v = 2:
       A[0, 2] = 1 and color[2] == 0 (unvisited).
       Update: pi[2] is set to 0.
       pi Changes to: [0, 0, 0, 1]
       dfs_visit(2) is called.
```

```
8. Entering dfs_visit(2):

    color[2] is set to 1 (visiting).
    time increments to 6.
    d[2] = 6.

    Exploring Neighbors of Node 2:

    Neighbor v = 0:
        A[2, 0] = 1 and color[0] == 2 (finished). No change to pi.
    Neighbor v = 1:
        A[2, 1] = 0 (no edge). No change to pi.
    Neighbor v = 2:
        A[2, 2] = 1 (self-loop), but color[2] == 1 (already visiting
            ). No change to pi.
    Neighbor v = 3:
        A[2, 3] = 1 and color[3] == 2 (finished). No change to pi.
    Post Exploration:
        color[2] is set to 2 (finished).
        time increments to 7.
        f[2] = 7.
        pi Remains: [0, 0, 0, 1]

9. Returning to dfs_visit(0):

    Neighbor v = 3:
        A[0, 3] = 1 and color[3] == 2 (finished). No change to pi.
    Post Exploration:
        color[0] is set to 2 (finished).
        time increments to 8.
        f[0] = 8.
        Final pi: [0, 0, 0, 1]

10. Continuing the DFS Loop:

    u = 1: color[1] == 2 (finished). No action taken.
    u = 2: color[2] == 2 (finished). No action taken.
    u = 3: color[3] == 2 (finished). No action taken.

Summary of pi Changes:

Trace | pi: [0, 1, 2, 3] -> [0, 0, 2, 3] -> [0, 0, 2, 1] -> [0, 0,
    0, 1]

So the output is:
{"output": "[[0, 1, 2, 3],[0, 0, 2, 3],[0, 0, 2, 1],[0, 0, 0, 1]]"}
```

---

## Prompts C.2: An Case from Codeforce in TracePile

<Problem> Vasya is very upset that many people on the Net mix uppercase and lowercase letters in one word. That's why he decided to invent an extension for his favorite browser that would change the letters' register in every word so that it either only consisted of lowercase letters or, vice versa, only of uppercase ones. At that as little as possible letters should be changed in the word. For example, the word HoUse must be replaced with house, and the word ViP — with VIP. If a word contains an equal number of uppercase and lowercase letters, you should replace all the letters with lowercase ones. For example, maTRIx should be replaced by matrix. Your task is to use the given method on one given word.
Input Specification:
The first line contains a word *s* — it consists of uppercase and lowercase Latin letters and possesses the length from 1 to 100.

Output Specification:
Print the corrected word *s*. If the given word *s* has strictly more uppercase letters, make the word written in the uppercase register, otherwise - in the lowercase one.

Demo Input:
{'HoUse', 'ViP', 'maTRIx'}

Demo Output:
{'house', 'VIP', 'matrix'}

Note:
none
<Problem>

And the input of the problem:
<Input> aOvSeu <Input>
**Predict the output of the problem by tracing the code step by step**:



```python
def solution(s):
    upper = sum(1 for c in s if c.isupper())
    lower = sum(1 for c in s if c.islower())
    if lower >= upper:
        return s.lower()
    else:
        return s.upper()
```



**CoE**:

Let's break down the execution of the provided code step by step for the input aOvSeu.

```
1. Input Processing:
    The input word is aOvSeu. It includes both uppercase and
        lowercase letters.

2. Counting Uppercase Letters:
Uppercase Letters Identified: O, S
Uppercase Count (upper): 2

3. Counting Lowercase Letters:
Lowercase Letters Identified: a, v, e, u
Lowercase Count (lower): 4

4. Decision Making Based on Counts:
Comparison: Is the number of lowercase letters (4) greater than or
    equal to the number of uppercase letters (2)?
Evaluation: 4 >= 2 -> True

5. Transforming the String:
Since the condition lower >= upper is True, the entire string is
    converted to lowercase.
Transformation:
    Original String: aOvSeu
    Transformed String: aovseu

6. Appending Newline Character:

Final Output: {"output": "aovseu"}.
```

**Prompts C.3: Case of the Biapartite Problem in TracePile**

**Input:** Given an undirected graph with nodes [Alice Johnson, Bob Smith, Catherine Lee, David Brown], the adjacency list is as follows: Alice Johnson: [Bob Smith, Catherine Lee, David Brown], Bob Smith: [Alice Johnson], Catherine Lee: [Alice Johnson], David Brown: [Alice Johnson]. The code with initialized parameters is as follows:
Code:

```python
def hungarian_algorithm_with_paths(adj_list, n_left, n_right_start):
    match_right = {}
    match_pairs = []

    def dfs(node, visited):
        for neighbor in adj_list.get(node, []):
            if not visited[neighbor]:
                visited[neighbor] = True
                if neighbor not in match_right:
                    match_right[neighbor] = node
                    return True
                elif dfs(match_right[neighbor], visited):
                    match_right[neighbor] = node
                    return True
        return False
    max_matching = 0
    for node in range(n_left):
        visited = {key: False for key in range(n_right_start,
            n_right_start + n_left)}
        if dfs(node, visited):
            max_matching += 1

    for right_node, left_node in match_right.items():
        match_pairs.append((left_node, right_node))

    return max_matching, match_pairs
```

What is the size of the maximum matching?
Please solve the problem by tracing through the code.
**CoE:**
Let's run the function hungarian_algorithm_with_paths(adj_list, n_left, n_right_start).

```
Initial values: adj_list = {Alice Johnson: [Bob Smith, Catherine Lee
    , David Brown], Bob Smith: [Alice Johnson], Catherine Lee: [
    Alice Johnson], David Brown: [Alice Johnson]}, n_left = Bob
    Smith, n_right_start = Bob Smith
Call hungarian_algorithm_with_paths({Alice Johnson: [Bob Smith,
    Catherine Lee, David Brown], Bob Smith: [Alice Johnson],
    Catherine Lee: [Alice Johnson], David Brown: [Alice Johnson]},
    Bob Smith,Bob Smith)
match_right = {} and match_pairs = [].
max_matching = 0
for node: Alice Johnson
Current match_right: {}
Initialize visited = {Bob Smith: False, Catherine Lee: False, David
    Brown: False}.
Call dfs(Alice Johnson,{Bob Smith: False, Catherine Lee: False,
    David Brown: False})
for neighbor: Bob Smith
Since neighbor Bob Smith is not visited
visited[Bob Smith] = True
match_right = {}
Since neighbor Bob Smith is not in match_right
match_right[Bob Smith] = Alice Johnson
Return True
Since dfs(Alice Johnson,{Bob Smith: True, Catherine Lee: False,
    David Brown: False}) returns True
```

```
max_matching = maximum_mactching + 1, max_matching: 1
Iterate over match_right: right_node = Bob Smith, left_node = Alice
    Johnson
Add pair (Alice Johnson,Bob Smith) to match_pairs
All nodes in set 1 are traversed, max_matching = [(Alice Johnson,
    Bob Smith)]
So the size of the maximum matching is 1

Final Output: {"output": "1"}.
```

## Prompts C.4: Case of mathematical problem from TracePile

Given a mathematical problem and its solution code, please analyze the execution of the above solution code step by step.
Provide a detailed, natural-language, step-by-step reasoning process that explains how the code solves the given mathematical problem.
Problem:
What is the sum of all the prime numbers between 1 and 10?

Solution Code:

```python
def is_prime(num):
    if num < 2:
        return False
    for i in range(2, int(num**0.5) + 1):
        if num % i == 0:
            return False
    return True

primes = [i for i in range(1, 11) if is_prime(i)]
print(sum(primes))
```

**CoE**:

```
1. **Define the 'is_prime' Function:**

   The function 'is_prime' is defined to check if a given number '
      num' is a prime number.
   - **Check if 'num' is less than 2:**
     - If 'num' is less than 2, the function returns 'False' because
         0 and 1 are not prime numbers.
   - **Check divisibility from 2 to the square root of 'num':**
     - The function iterates through all numbers from 2 to the
         integer part of the square root of 'num' (inclusive).
     - For each number 'i' in this range, it checks if 'num' is
         divisible by 'i' (i.e., 'num % i == 0').
     - If 'num' is divisible by any 'i', it returns 'False' because
         'num' is not a prime number.
   - **Return 'True' if no divisors are found:**
     - If the loop completes without finding any divisors, the
         function returns 'True', indicating that 'num' is a prime
         number.

2. **Generate the List of Prime Numbers:**

   - A list comprehension is used to generate a list of prime
       numbers between 1 and 10.
   - The list comprehension iterates through each number 'i' in the
       range from 1 to 10 (inclusive).
   - For each 'i', it calls the 'is_prime' function.
   - If 'is_prime(i)' returns 'True', the number 'i' is included in
       the list 'primes'.
```

3. **Calculate the Sum of Prime Numbers:**

    - The `sum` function `is` used to calculate the `sum` of `all` the
      numbers `in` the `list` `primes`.

4. **Output the Result:**

    - The result of the `sum` function `is` printed to the console.

### Detailed Execution

1. **Check `if` 1 `is` a prime number:**
    - `is_prime(1)`: Since 1 `is` less than 2, the function returns `
      False`.
    - 1 `is` `not` included `in` the `list` `primes`.

2. **Check `if` 2 `is` a prime number:**
    - `is_prime(2)`: 2 `is` `not` less than 2.
    - The loop `from` 2 to the square root of 2 (which `is` 1.414, so the
      integer part `is` 1) `is` `not` executed because the `range` `is`
      empty.
    - The function returns `True`.
    - 2 `is` included `in` the `list` `primes`.

3. **Check `if` 3 `is` a prime number:**
    - `is_prime(3)`: 3 `is` `not` less than 2.
    - The loop `from` 2 to the square root of 3 (which `is` 1.732, so the
      integer part `is` 1) `is` `not` executed because the `range` `is`
      empty.
    - The function returns `True`.
    - 3 `is` included `in` the `list` `primes`.

4. **Check `if` 4 `is` a prime number:**
    - `is_prime(4)`: 4 `is` `not` less than 2.
    - The loop `from` 2 to the square root of 4 (which `is` 2) checks `if`
      4 `is` divisible by 2.
    - Since 4 `is` divisible by 2, the function returns `False`.
    - 4 `is` `not` included `in` the `list` `primes`.

5. **Check `if` 5 `is` a prime number:**
    - `is_prime(5)`: 5 `is` `not` less than 2.
    - The loop `from` 2 to the square root of 5 (which `is` 2.236, so the
      integer part `is` 2) checks `if` 5 `is` divisible by 2.
    - Since 5 `is` `not` divisible by 2, the function returns `True`.
    - 5 `is` included `in` the `list` `primes`.

6. **Check `if` 6 `is` a prime number:**
    - `is_prime(6)`: 6 `is` `not` less than 2.
    - The loop `from` 2 to the square root of 6 (which `is` 2.449, so the
      integer part `is` 2) checks `if` 6 `is` divisible by 2.
    - Since 6 `is` divisible by 2, the function returns `False`.
    - 6 `is` `not` included `in` the `list` `primes`.

7. **Check `if` 7 `is` a prime number:**
    - `is_prime(7)`: 7 `is` `not` less than 2.
    - The loop `from` 2 to the square root of 7 (which `is` 2.646, so the
      integer part `is` 2) checks `if` 7 `is` divisible by 2.
    - Since 7 `is` `not` divisible by 2, the function returns `True`.
    - 7 `is` included `in` the `list` `primes`.

8. **Check `if` 8 `is` a prime number:**
    - `is_prime(8)`: 8 `is` `not` less than 2.

- The loop `from` 2 to the square root of 8 (which `is` 2.828, so the
          integer part `is` 2) checks `if` 8 `is` divisible by 2.
        - Since 8 `is` divisible by 2, the function returns `False`.
        - 8 `is not` included `in` the `list` `primes`.

9.  **Check `if` 9 `is` a prime number:**
        - `is_prime(9)`: 9 `is not` less than 2.
        - The loop `from` 2 to the square root of 9 (which `is` 3) checks `if`
          9 `is` divisible by 2 `and` 3.
        - Since 9 `is` divisible by 3, the function returns `False`.
        - 9 `is not` included `in` the `list` `primes`.

10. **Check `if` 10 `is` a prime number:**
        - `is_prime(10)`: 10 `is not` less than 2.
        - The loop `from` 2 to the square root of 10 (which `is` 3.162, so
          the integer part `is` 3) checks `if` 10 `is` divisible by 2 `and` 3.
        - Since 10 `is` divisible by 2, the function returns `False`.
        - 10 `is not` included `in` the `list` `primes`.

### Final List of Primes

The `list` `primes` contains the numbers: [2, 3, 5, 7].

### Calculate the Sum

- The `sum` of the `list` `primes` `is` calculated: 2 + 3 + 5 + 7 = 17.

### Output the Result

- The final output `is` 17.

### Final Answer
17

