# OpenReview forum: "Chain of Execution Supervision Promotes General Reasoning in Large Language Models"
_NeurIPS.cc/2025/Conference — NeurIPS 2025 poster_

### Official Review · Reviewer_fNHA · 2025-06-24

**Clarity:** 3
**Significance:** 2
**Originality:** 2
**Rating:** 4
**Confidence:** 5

**Summary:**

This paper introduces TracePile, a corpus of 2.6 million samples containing 25 billion tokens that transforms code execution into natural language COE explanations. The authors collect code from algorithmic competitions (Codeforces), classical algorithms (CLRS), and mathematical problems (OpenMath), then use Qwen-2.5-72B-Instruct to generate step-by-step execution traces. They evaluate this approach across three training paradigms (continue-pretraining, instruction tuning after pretraining, and two-stage finetuning) on four base models (LLaMA 3/3.1, Qwen-2.5, Qwen-2.5 Coder), reporting improvements on 20 benchmarks spanning mathematics, code, logic, and algorithms.

**Questions:**

Please see my comments above.

In addition, while authors mentioned that "we impose a strict 8k-token limit", from Table 2 it seems the avg #tokens is greater than 8k?

**Ethical Concerns:**

["NO or VERY MINOR ethics concerns only"]

**Final Justification:**

Thanks authors for the updates. These address my concerns and I have thus raised my score to 4. Please make sure these discussions appear in the final version of the paper.

**Limitations:**

See "Strengths And Weaknesses"

**Quality:**

3

**Strengths And Weaknesses:**

The paper addresses a compelling hypothesis that explicit execution traces can teach transferable reasoning skills. The scale of the effort (2.6M samples, 25B tokens) is substantial, and the diversity of sources (competitions, textbook algorithms, mathematical problems) provides good coverage of different reasoning patterns. The evaluation offers useful insights about integration strategies.

However, several critical weaknesses undermine the contribution. Most significantly, the baseline comparisons appear unfair. The authors compare against CodeI/O [25] using different training configurations, e.g. CodeI/O uses only 1 epoch while TracePile uses 3 epochs in first stage SFT, this invalidates the claim and conclusion drawn in the paper e.g. RQ1/Table 4.

The paper also keeps comparing models fine-tuned on TracePile against base models which tells us nothing about the relative value of CoE versus other supervision approaches. A fair comparison would be CoE against models trained on equivalent amounts of alternative data (e.g., more code examples, mathematical solutions, or even generic instruction data) using identical hyperparameters.

The related work section is missing in the main paper which is not acceptable. In addition, there is no discussion on how this work is situated in the rich past literature on using execution traces to train models, e.g., TRACED, SemCoder, and CodeT. It'd be great if the authors could clearly state the novelty beyond existing work.

The intrinstic evaluation is also missing from the paper, i.e., whether the model learns to generate *correct* traces. It'd be a good add-on to complement existing evaluation.

---

> ### Author Rebuttal · Authors · 2025-07-29
>
> We sincerely thank you for the comprehensive review and the constructive feedback shared. Below are our responses to your insightful comments.
> ## Q1: Unfair comparison with CodeIO
> Thank you for your valuable feedback regarding the baseline comparison. We understand your concern about the differing training epochs (1 for CodeI/O vs. 3 for TracePile).
>
> Firstly, we understand your emphasis on consistent training configurations. However, it's important to note that **due to significant differences in the scale and characteristics of our TraceMind dataset compared to the dataset used by CodeI/O, directly enforcing the same training epochs might not fully unleash the potential of our method**. Different datasets often necessitate varying training strategies and convergence steps to achieve optimal performance.
>
> Crucially, to ensure fairness, we also evaluated our model after just 1 epoch of training. Even in this configuration, our model significantly outperformed CodeI/O, demonstrating the inherent effectiveness of our approach and the TraceMind dataset. We will clarify these details and consider adding these comparative results to the paper to fully address your point.
>
> |Models (Qwen-2.5-Code)|GSM8K|MATH|Livecodebench-Output|Zebra logic|
> |---|---|---|---|---|
> |CodeIO|79.3|39.8|17.6|8.6|
> |Ours-1epoch|80.2|47.7|30.5|9.4|
> |Ours-3epoch |80.8|50.8|35.5|10.8|
>
> In our experiments, we indeed conducted extensive explorations of model performance across different training epochs. The experimental results consistently showed that within 3 epochs, the model's performance continued to improve with more training steps, exhibiting superior average performance. Therefore, to present the best performance of TracePile in the paper, we chose to report the results obtained after 3 epochs of training.
>
> ## Q2: Comparison with different supervision approaches
> Thank you for your crucial feedback regarding the fairness and scope of our baseline comparisons. We entirely agree that ensuring fair training and testing conditions is paramount to accurately demonstrate the unique value of Chain of Execution (CoE) supervision against other approaches.
>
> We agree that fair comparisons against alternative supervision methods are crucial.  Our paper already includes extensive comparisons beyond base models, as detailed in our discussion (RQ1) and Tables 3, 4, and 7.
>
> - Generic Instruction Data: Tables 3 and 7 show comparisons against TuluSFT, a widely recognized generic instruction dataset.
>
> - Alternative Structured Code Data: Table 4 explicitly compares TracePile against OpenMath (pure code solution data) and CodeI/O (different forms of code input-output data).
>
> To provide a clearer overview of these comparisons,  we have conducted a comprehensive set of experiments using the Qwen-2.5-Coder 7B model under a two-stage fine-tuning paradigm. This allows for a more direct and equitable comparison of TracePile against various alternative supervision approaches. For OpenMathInstruct 1 and 2, we specifically selected 2.6M subsets and trained for 3 epochs to ensure comparable data scale and training duration.
>
> |Models (Qwen-2.5-Coder)|Stage-1|Stage-2|GSM8K|MATH|Livecodebench-Output|Zebra logic|
> |---|---|---|---|---|---|---|
> |Baseline|-| Tulusft|74.3|45.0|29.8|9.2|
> |Pure Code Solution| OpenmathInstruct 1-2.6M (3 epoch) |Tulusft| 80.1 | 46.1 |19.4 | 5.6|
> |Pure Math Solution| OpenmathInstruct 2-2.6M (3 epoch) |Tulusft| 82.1 | 51.1 |17.4 | 6.6|
> |generic instruction|Webinnstruct-2.6M (3 epoch)| Tulusft|81.3|49.0|18.8|9.1|
> |CodeIO | CodeIO-3.5M| Tulusft |79.3|39.8|17.6|8.6|
> |Ours-1epoch|Tracepile-2.6M (1epoch)|Tulusft|80.2|47.7|26.5|9.4|
> |Ours-3epoch|Tracepile-2.6M (1epoch)|Tulusft|80.8|50.8|35.5|10.8|
>
> As shown, TracePile consistently excels across diverse reasoning domains. While pure math solutions perform slightly better on math benchmarks due to specialization, TracePile significantly outperforms all other baselines, particularly in LiveCodeBench-Output and Zebra Logic. This highlights TracePile's ability to enhance **generalizable reasoning capabilities** by fostering transferable understanding of logical and procedural patterns.
>
> ## Q3: Concerns about Related Work
> We apologize for the related work section's placement. It is indeed included in Appendix A (lines 536-562) due to space limitations. We will explicitly reference it in the main paper.
>
> Our work introduces fundamental and distinct contributions beyond prior literature like TRACED, SemCoder, and CodeI/O:
>
> 1.  **Execution Tracing Granularity:**
>     * **TRACED:** While TRACED also uses code traces, its backbone is RoBERTa, limiting it to simpler classification tasks and primarily focusing on code understanding. **Crucially, its traces are not in natural language, which restricts its applicability to general reasoning tasks in LLMs.**
>     * **SemCoder:** Employs "monologue reasoning" with high-level functional descriptions and verbal reasoning about local execution effects. However, it primarily uses synthetic data with limited real-world complexity.
>     * **CodeI/O:** Condenses reasoning patterns by predicting inputs/outputs from given functions in natural language, but its explanations focus predominantly on abstract input-output reasoning without detailed execution tracing of internal program states.
>     * **TracePile (Ours):** Explicitly emphasizes fine-grained, structured execution traces (Chain-of-Execution, CoE), capturing detailed variable state changes and control flow at each step in natural language. This structured granularity significantly surpasses the abstract verbal reasoning of SemCoder and CodeI/O, enabling models trained on TracePile to exhibit deeper understanding and robust execution reasoning capabilities.
>
> 2.  **Data Sources:**
>     * **SemCoder:** Relies entirely on synthetic data (PYX).
>     * **CodeI/O:** Derives data from generic raw code files transformed into structured reasoning tasks.
>     * **TracePile (Ours):** Uniquely integrates diverse, real-world-oriented sources, including:
>         * Classical algorithms with intermediate checks.
>         * Algorithmic competition code verified with detailed execution traces.
>         * Synthetic algorithmic problems designed for high complexity.
>         * Mathematical code problems for multi-step decomposition.
>
> 3.  **Training Paradigm:**
>     * **SemCoder and CodeI/O:** Primarily rely on instruction tuning for explanations.
>     * **TracePile (Ours):** Adopts three different multi-stage training paradigms: **continue pretraining (CPT) + few-shot prompting**; **continue pretraining + general instruction tuning**; and **two-stage instruction tuning**. We are the first to demonstrate that CoE-similar format data can significantly contribute to model performance in the CPT stage. Our comprehensive training settings further push the boundaries of using this type of data in the field.
>
> We believe that these distinctions, particularly our emphasis on fine-grained, natural language CoE traces from diverse real-world sources and our exploration of multi-stage training paradigms, clearly establish the novelty and significant contribution of TracePile to the literature on enhancing LLM reasoning through execution supervision. We will ensure these points are more prominently highlighted in the main paper to address your concern directly.
>
> ## Q4: Intrinsic Evaluation
> Thank you for highlighting the importance of intrinsic evaluation, specifically whether our model learns to generate correct traces. We agree that this is a crucial aspect to complement existing evaluations and directly demonstrate the value of our Chain of Execution (CoE) supervision.
>
> To address this, we conducted an additional, insightful experiment. We compared our TracePile CoE method against baseline approaches, including CodeI/O, on the LiveCodeBench-Output test set. Beyond just measuring the final output accuracy, we also assessed the** accuracy of the intermediate steps** generated by the models. For this intermediate step accuracy verification, we leveraged a powerful external model, **Qwen3-235B-A22B-Instruct-2507**, to check the correctness of each step in the generated traces.
>
> Here are the results of this intrinsic evaluation:
> |Models (Qwen-2.5-Coder)|Final Output Accuracy (%)|Intermediate Step Accuracy (%)|
> |---|---|---|
> |Baseline (TuluSFT)|27.2|18.8|
> |CodeI/O++|20.6|17.1|
> |Our Method (TracePile CoE)|**35.5**|**33.7**|
>
> **Intermediate Step Accuracy Explanation**: *This metric represents the proportion of samples where the final output is correct AND all generated intermediate reasoning steps are also verified as correct.* This metric more rigorously measures the end-to-end reasoning fidelity of the model.
>
> As the table demonstrates, our TracePile CoE method not only achieves superior final output accuracy on LiveCodeBench-Output but also shows a significantly higher accuracy in its intermediate steps compared to both the generic instruction baseline and CodeI/O++.
>
> We will adopt this more precise definition and ensure its calculation method is clearly articulated in the final version of the paper to more comprehensively demonstrate the advantages of our approach. Thank you again for your invaluable suggestion!
>
>
> ## Q5: Average token of Tracepile
>
> Regarding your question about the 8k-token limit and the average token count in Table 2, you are correct to notice a potential discrepancy.
>
> Upon re-examination, we found Table 2's total token count inadvertently included ***tokens from few-shot examples within the data generation prompts***. After removing these, the corrected average token length per training sample is approximately **7452** tokens, and the total TracePile token count is around **19.5** billion tokens. This revised statistic accurately reflects the data adhering to our 8k-token limit and resolves the apparent contradiction. We will update Table 2 accordingly.

---

> > ### Author Response · Authors · 2025-08-04
> > **Rebuttal Review Request**
> >
> > Dear Reviewer,
> >
> > Thank you again for taking the time to review our paper and for providing valuable feedback. We have done our best to address your concerns in the rebuttal.
> >
> > We would greatly appreciate it if you could take a moment to review our responses and let us know if any concerns remain.
> >
> > Thank you once again for your thoughtful review.
> >
> >
> > Additionally, we noticed a small typo in one of our tables after response submission. Specifically, in the **table of Q2**, the value of last row “Tracepile-2.6M (1epoch)” should be “Tracepile-2.6M (3epoch)”. This typo does not affect any conclusions or claims, and we sincerely apologize for the oversight.
> >
> > The Authors

---

> ### Comment · Reviewer_fNHA · 2025-08-06
>
> Dear AC,
>
> Thanks for your message. I had updated the final justification in my main review as instructed by the new policy. I don’t have further questions for the authors.
>
> Best,
>
> Reviewer

---

### Official Review · Reviewer_ruAL · 2025-07-02

**Clarity:** 3
**Significance:** 3
**Originality:** 3
**Rating:** 5
**Confidence:** 2

**Summary:**

The paper presents a technique to augment data for continued pertaining of large language models to improve the coding and math abilities of the models. The technique constructs a dataset TracePile by collecting algorithmic data of 3 sources: competition code (mainly from Codeforces), classical algorithmic code, and mathematical code. These collected samples are further diversified by modifying the query or the code. The technique uses a language model (Qwen-2.5-72B in this case) to obtain the chain-of-execution data from under carefully designed prompting strategies.

The TracePile dataset is used to continually pretrain 4 base models: Llama 3 & 3.1, Qwen-2.5, and Qwen-2.5. The continue-pretrained model shows consistent improvement over the base model on a diverse set of coding and math problems.

**Questions:**

How do the experiments avoid test-data contamination?

**Ethical Concerns:**

["NO or VERY MINOR ethics concerns only"]

**Final Justification:**

Authors have addressed my concerns.

**Limitations:**

Yes

**Quality:**

3

**Strengths And Weaknesses:**

**Strenghts:**

1. The paper is well-written and easy to follow. The idea is simple and effective.

2. Experiments are comprehensive and cover diverse coding and math benchmarks.

3. The research questions discussed in Section 4 are insightful and cover most of the crucial ablations. The RQ1 shows that chain-of-execution data is more effective than I/O style supervision from previous work, CodeI/O (Li et. al. 2025).

**Weaknesses:**

1. The experiments are focused on coding and mathematical benchmarks, and authors should show the effect of proposed continual pretraining on other common benchmarks not related to coding or math.

2. The code or data is not shared. Justification for the checklist question 5 about code and data is incomplete. Some of the other answers to the checklist questions are unclear as well.

**Minor:**
Typo: LLaMA -> Llama

---

> ### Author Rebuttal · Authors · 2025-07-29
>
> We are truly grateful for your thorough review and insightful feedback. Our responses to your comments are outlined below.
>
>
> ## Q1: Effects of CoE in other reasoning benchmarks beyond math and code.
>
> Thank you for requesting further clarification on the effect of our proposed continual pretraining on benchmarks not related to coding or mathematics. We appreciate this opportunity to highlight the broader generalizability of our method. As you noted, this information is indeed already present in our paper.
>
> Specifically, in Table 1 of our submission, we report the performance improvements of models after the continual pretraining stage across various general reasoning benchmarks, including logical and algorithmic reasoning.
>
> Below is a summary table, extracted from our paper, showcasing the performance of Llama-3-8B and Llama-3.1-8B models on these non-coding and non-mathematical benchmarks after continual pretraining with TracePile:
>
> | Model         | Domain              | Dataset        | Baseline (Base) | Our Method (Ours) | Improvement (%) |
> |---------------|---------------------|----------------|------------------|--------------------|------------------|
> | LLaMA-3-8B     | Logical Reasoning   | Zebra Puzzle   | 0.1              | 2.8                | **+2.7**             |
> |               |                     | KORBench       | 7.4              | 21.8               | **+14.4**            |
> |               |                     | Ruletaker      | 21.8             | 42.4               | **+20.6**            |
> |               | Algorithm Reasoning | Graphwiz       | 4.5              | 46.0               | **+41.5**            |
> |               |                     | GraphInstruct  | 35.2             | 73.5               | **+38.3**            |
> |               |  Instruction-Following                   | IFEval  | 11.5             | 25.5               | **+14.0**            |
> | LLaMA-3.1-8B   | Logical Reasoning   | Zebra Puzzle   | 2.3              | 20.6               | **+18.3**            |
> |               |                     | KORBench       | 7.4              | 45.6               | **+38.2**            |
> |               |                     | Ruletaker      | 21.9             | 45.6               | **+23.7**            |
> |               | Algorithm Reasoning | Graphwiz       | 33.9             | 69.2               | **+35.3**            |
> |               |                     | GraphInstruct  | 33.0             | 69.2               | **+36.2**            |
> |               |  Instruction-Following                   | IFEval  | 11.6             | 21.3              | **+9.7**            |
>
> As the table illustrates, our TracePile method, during the continual pretraining stage, yields substantial performance gains on Llama 3-8B and Llama 3.1-8B models across logical and algorithmic reasoning  and instruction following tasks.
>
>
> ## Q2: Access to Data
> Thank you for your diligent review and for pointing out critical issues regarding our NeurIPS checklist responses, particularly concerning the transparency of code and data access. We apologize for any ambiguity or incompleteness in our initial submission.
>
> You are absolutely correct that our response to Checklist Question 5 ("Open access to data and code") is misleading if the code and data are not openly shared. We acknowledge this discrepancy.
>
> Indeed, TracePile was not publicly released at submission due to **company-related copyright constraints**. As per NeurIPS guidelines, we are unable to introduce new data or anonymous links during the rebuttal phase. Currently, we are actively processing internal company approval for public release of TracePile in one month. Once approved, we commit to releasing the full dataset on Hugging Face, providing comprehensive instructions and scripts to ensure full reproducibility. ***This ongoing approval process is also why we have refrained from publicly posting our manuscript to platforms such as arXiv at this time.***
>
> ## Q3: Avoid test-data contamination
>
> Thank you for your critical question regarding test-data contamination. Ensuring the validity and trustworthiness of experimental results is paramount.
>
> **How Test-Data Contamination is Avoided (General Principles):**
>
> Avoiding test-data contamination primarily involves:
> 1.  **Strict Data Splitting:** Ensuring no overlap between training, validation, and test sets. Test data must remain unseen during all training and hyperparameter tuning phases.
> 2.  **Independent Data Sources:** Ideally, test data should originate from sources entirely separate from training data, or represent novel scenarios the model has not encountered.
> 3.  **Leveraging Established Benchmarks:** Utilizing widely accepted public benchmarks with pre-defined splits and robust de-duplication/anti-contamination measures.
> 4.  **Preventing Information Leakage:** Ensuring no implicit information from the test set influences the training process (e.g., through feature engineering, data augmentation, or model selection that "peeks" at test data).
>
> **How We Avoided Test-Data Contamination in Our Paper:**
>
> In our work, we implemented several rigorous measures to prevent test-data contamination and ensure the validity of our experimental results:
>
> 1.  **Independent Evaluation Benchmarks:** We conducted comprehensive evaluations on **20 diverse, publicly available benchmarks** spanning mathematics, code, logical, and algorithmic reasoning (as detailed in **Section 3.1** and **Appendix C** of our paper). These benchmarks (e.g., GSM8K, MATH, LiveCodeBench, CRUX, Zebra Logic) are well-established with pre-defined splits and are managed by public evaluation toolkits (OpenCompass, Qwen2.5-Math, ZeroEval), ensuring consistency and reproducibility.
>
> 2.  **Domain and Format Discrepancy:** Our TracePile dataset, while code-execution-based, focuses on **fine-grained, CoE-style natural language reasoning traces**. The evaluation benchmarks, especially those targeting out-of-domain generalization (e.g., LiveCodeBench, CRUX, RuleTaker, Zebra Logic), often differ significantly in format, domain, or reasoning style from our training sources (as discussed in **Section 4, "Out-of-domain (OOD) Generalization"**). This distinction helps ensure that the model learns generalizable reasoning capabilities rather than simply memorizing test patterns.
>
> 3.  **Rigorous Data Generation and Filtering:**
>     * For mathematical code data, we employed a **model-based filtering strategy** (Section 2.1) where problems correctly answered three times by LLaMA3-8B were discarded as too simple, preventing overfitting to trivial or potentially similar test patterns.
>     * We enforced **intermediate result verification** and **consistency checks** (Section 2.3) for our CoE data, ensuring logical rigor beyond just the final answer. This encourages the model to learn deep reasoning logic.
> These measures collectively ensure the reliability of our experimental results and the generalizability of our model's enhanced reasoning capabilities.

---

> > ### Comment · Reviewer_ruAL · 2025-08-05
> >
> > Authors have addressed my concerns and I have raised my score accordingly.

---

> > > ### Author Response · Authors · 2025-08-06
> > >
> > > Thank you for your valuable feedback and for raising your score. We truly appreciate your time and thoughtful review of our paper.

---

### Official Review · Reviewer_2Sy7 · 2025-07-03

**Clarity:** 3
**Significance:** 3
**Originality:** 2
**Rating:** 4
**Confidence:** 4

**Summary:**

This paper introduces TracePile, a data corpus of 2.6 million samples designed to enhance reasoning abilities in large language models through Chain of Execution (CoE) supervision. This paper proposes to transform code execution into explicit, step-by-step natural language explanations that trace through mathematical problems, classical algorithms, and algorithmic competition code. The dataset spans diverse reasoning paradigms with additional augmentation strategies for query and code diversification to enhance logical granularity. Through experiments across four base models (LLaMA 3/3.1 and Qwen-2.5) and 20 benchmarks covering mathematical, logical, algorithmic, and code reasoning tasks, the authors demonstrate that TracePile consistently improves performance over these benchmarks.

**Questions:**

- Could the authors add a discussion section in the paper to explain how their approach is similar to and different from SemCoder and Code I/O? It is ok to suggest CoE is inspired by existing works, but a discussion to specify the similarity and difference will be helpful to understand the unique contribution of this work
- Could the authors submit the data corpus for reviewers to assess its quality during the rebuttal phase?

**Ethical Concerns:**

["NO or VERY MINOR ethics concerns only"]

**Final Justification:**

After the communication during the rebuttal, I keep my score of 4 as my final attitude

I appreciate the authors commitment of adding more detailed discussion and comparison with SemCoder and Code I/O in the camera-ready.

However, though I understand the policy from certain companies that the publication of data and code needs approval, I still choose to be conservative about my rating at this time -- I keep my score of 4 to show my positive attitude toward this paper and the authors' commitment regarding the final open-source artifacts, but without manually assessing the quality of TracePile, I choose not to increase my score to an absolute level (i.e., 5) at this moment.

**Limitations:**

Yes

**Paper Formatting Concerns:**

No significant formatting concern observed.

**Quality:**

2

**Strengths And Weaknesses:**

## Strengths

+ The paper provides a large corpus to learn the code execution reasoning, which successfully improves LLMs' general reasoning capabilities -- this successful knowledge transfer is and exciting finding.
+ This paper implements "intermediate results verification" to ensure the quality of the annotated reasoning text, which guarantees the correctness of intermediate reasoning steps.
+ Training with TracePile consistently boosts base models of varied sizes with better reasoning performance across multiple reasoning benchmarks.

## Weaknesses

- Limited Technical Contributions Over Existing Works

While TracePile presents a large-scale dataset for code execution supervision, the paper's core contribution exhibits limited novelty compared to recent work in code execution reasoning, particularly SemCoder (Ding et al., 2024) and CodeI/O (Li et al., 2025). SemCoder already introduced the concept of training code language models with comprehensive semantics reasoning through "monologue reasoning," where models learn to articulate code execution step-by-step in natural language, similar approach to TracePile's Chain of Execution (CoE). Similarly, CodeI/O also transforms code execution into natural language explanations for training purposes. While TracePile claims to focus on "fine-grained execution traces," this distinction appears incremental rather than fundamentally novel, as existing works already demonstrated that natural language execution reasoning could effectively bridge static code and dynamic execution states.

- Unavailability of the Core Dataset and Inadequate Reproducibility Provisions

A fundamental weakness of this paper is the unavailability of TracePile, the core 2.6 million sample dataset that constitutes the primary contribution. Despite claiming that their work introduces a "large-scale corpus" as the main contribution, the authors fail to provide access to the actual dataset, severely limiting the paper's impact and reproducibility (Please correct me if I am wrong).

Furthermore, the authors' response to the reproducibility checklist is misleading—they cite "Section 3.1: Experimental Settings" as evidence for providing "open access to data and code with sufficient instructions to faithfully reproduce the main experimental results." However, Section 3.1 only describes training hyperparameters and evaluation protocols, not data availability or access instructions. The experimental settings alone are insufficient for reproducibility when the core dataset remains inaccessible.

For a paper where the dataset is the primary contribution and the foundation for all experimental results, the failure to provide data access represents a critical flaw that undermines the work's scientific value. The authors should commit to releasing TracePile during the rebuttal phase to enable proper peer review and future research validation.

- (minor) In the abstract, the data corpus was named as TraceMind at first. I assume this should be a typo

---

> ### Author Rebuttal · Authors · 2025-07-29
>
> We sincerely thank you for the comprehensive review and the constructive feedback shared. Below are our concise responses to your insightful comments.
> ## Q1: Concerns about the novelty and incremental nature of TracePile compared to existing works (SemCoder and CodeI/O).
>
> We appreciate the reviewer for highlighting the critical aspect of novelty and differentiation. Although TracePile shares the general goal of leveraging natural language for code execution reasoning with SemCoder  and CodeI/O, our work introduces fundamental and distinct contributions:
>
> ### Execution Tracing Granularity:
>
> - SemCoder employs "monologue reasoning," emphasizing high-level functional descriptions, constraints, and verbal reasoning about local execution effects. However, it primarily uses synthetic data generated from scratch with limited real-world complexity
> - CodeI/O condenses reasoning patterns by predicting inputs/outputs from given functions in natural language, but the explanations focus predominantly on abstract input-output reasoning without detailed execution tracing of internal program states
> - TracePile (ours) explicitly emphasizes fine-grained, structured execution traces (Chain-of-Execution, CoE), capturing detailed variable state changes and control flow at each step. This structured granularity significantly surpasses the abstract verbal reasoning of SemCoder and CodeI/O, enabling models trained on TracePile to exhibit deeper understanding and robust execution reasoning capabilities.
> ### Data Sources:
>
> While SemCoder’s data (PYX) is entirely synthetic and CodeI/O derives data from generic raw code files transformed into structured reasoning tasks, TracePile uniquely integrates  diverse, real-world-oriented sources: *classical algorithms with intermediate checks, algorithmic competition code verified with detailed execution traces, and synthetic algorithmic problems designed for high complexity，and mathematical code problems for  multi-step decomposition*.
>
> This deliberate selection ensures broad coverage of real-world coding challenges, including tasks that demand intensive structural reasoning, dynamic tracking of complex data structures (arrays, stacks, heaps), and control-flow interpretation—scenarios less comprehensively addressed by SemCoder or CodeI/O.
> ### Training Paradigm:
>
> SemCoder and CodeI/O primarily rely on **instruction tuning** explanations. TracePile adopts three different  multi-stage training paradigm: **continue pretraining (cpt) + few-shot prompting**; **continue pretraining + general instruction tuning**; **two-stage instruction tuning**. We are the first to prove that CoE-similar format data could contribute the model performancnes in cpt stage. Our compresive trainning settings further push the boundry of using data in the field.
>
>
> In summary, TracePile distinguishes itself clearly through its structured, fine-grained trace annotation, comprehensive use of real-world coding scenarios with detailed instrumentation, and multi-stage augmented training strategy, thus substantially enhancing model performance on complex, real-world execution reasoning tasks.
>
> We will clarify these differences explicitly in the revised manuscript.
>
> ## Q2: Concerns about data availability and reproducibility due to the non-release of the TracePile dataset.
>
>  We sincerely appreciate the reviewer highlighting the critical aspect of data availability and reproducibility. Indeed, TracePile was not publicly released at submission due to company-related copyright constraints. As per NeurIPS guidelines, we are unable to introduce new data or anonymous links during the rebuttal phase. Currently, we are actively processing internal company approval for public release of TracePile in one month. Once approved, we commit to releasing the full dataset on Hugging Face, providing comprehensive instructions and scripts to ensure full reproducibility. ***This ongoing approval process is also why we have refrained from publicly posting our manuscript to platforms such as arXiv at this time***.
>
> ## Q3: Suggestion to add a discussion explicitly comparing TracePile to SemCoder and CodeI/O, clearly highlighting similarities and differences.
>
> We thank the reviewer for this excellent suggestion, which indeed helps clarify our unique contributions. We partially addressed this comparison in **our related work section** (e.g., lines 552–562 for CodeI/O), highlighting some key differences. To further enhance clarity, we will expand the discussion explicitly comparing TracePile's Chain-of-Execution (CoE) to SemCoder and CodeI/O, clearly stating how our structured fine-grained execution traces, real-world-oriented datasets, and multi-stage augmentation differ from these prior works (**R1 to Q1**). We agree that explicitly specifying both similarities and differences will greatly benefit the reader’s understanding of our unique contributions, and we will incorporate this detailed discussion into the revised manuscript.

---

> > ### Comment · Reviewer_2Sy7 · 2025-08-07
> >
> > Thanks for the authors' responses.
> >
> > I appreciate the authors commitment of adding more detailed discussion and comparison with SemCoder and Code I/O in the camera-ready.
> >
> > However, though I understand the policy from certain companies that the publication of data and code needs approval, I still choose to be conservative about my rating at this time -- I keep my score of 4 to show my positive attitude toward this paper and the authors' commitment regarding the final open-source artifacts, but without manually assessing the quality of TracePile, I choose not to increase my score to an absolute level (i.e., 5) at this moment.

---

> > > ### Author Response · Authors · 2025-08-07
> > >
> > > Dear Reviewer,
> > >
> > > Thank you for your thoughtful response and for maintaining your positive score. We deeply appreciate your understanding regarding the importance of assessing TracePile's quality through open-source artifacts.
> > >
> > > We reiterate our strong commitment to open-sourcing TracePile and its code upon acceptance, pending internal company approval processes. We hope our detailed clarifications and this commitment sufficiently address your concerns for this stage.
> > >
> > > Thank you again for your valuable insights.

---

### Official Review · Reviewer_R7E8 · 2025-07-14

**Clarity:** 3
**Significance:** 3
**Originality:** 3
**Rating:** 5
**Confidence:** 3

**Summary:**

- This work presents TracePile, a large dataset of "chain of execution" (CoE) rationales, which are natural language chain of thought explanations of code execution.
- Their code comes from 3 sources: (1) they use algorithmic competition code (primarily Codeforces) and prompt an LLM to extract the functions and reference input/output examples, verifying that the code executes to the actual input and output. (2) For a set of 30 classic algorithms, they use generators from prior work (CLRS-text and GraphInstruct) to generate intermediate states which they can use to verify intermediate results of chain of execution. (3) They also add Math code from OpenMath which is already paired with solver code, and filter out any problems that are solved by LLaMA3-8B in all three of three tries.
- For the algorithmic and graph problems, they diversify the dataset by asking not only about output prediction but also to predict intermediate values – an LLM generates these variants. And furthermore they diversify the code by having an LLM rewrite the code to use different approaches to the same problem or to be refactored differently.
- Finally they few-shot prompt Qwen-2.5-72B-Instruct to generate the actual CoE traces, either telling it to answer the given question by tracing code execution or telling it to traces the changes to a particular variable while explaining each step. They verify these intermediate states for the algorithmic problems using ground truth traces.
- They explore continue-pretraining on TracePile, pretraining on TracePile then instruction tuning on TuluSFT, and fine tuning first on TracePile then on TuluSFT; the last of these options provides the largest benefit. Training boosts performance considerably on coding and math tasks, but also other domains outside the dataset like logical reasoning.

**Questions:**

- See questions in "Weaknesses"
- Why was the particular set of benchmarks chosen for table 4?
- Lines 306-308 state: "Among them, code rewrites contribute most to structural generalization, while query diversification enhances task-specific interpretability." Unlike the other lines of this paragraph, this one didn't obviously follow from the results in the table for me. Where does this conclusion come from?
- Lines 154-155 state: "For tasks requiring variable tracking or control-flow interpretation, we require the model to output structured json results." What is meant by "control flow interpretation" here? If it's any tasks requiring control flow like loops or if-statements that must be almost every coding task?

**Ethical Concerns:**

["NO or VERY MINOR ethics concerns only"]

**Final Justification:**

Questions and concerns were resolved, maintaining a score of Accept as justified by my original review.

**Limitations:**

Yes

**Quality:**

3

**Strengths And Weaknesses:**

I've identified a few minor issues and points of confusion, but overall I believe this paper is a good fit for NeurIPS and support its acceptance. The evaluation is very thorough and results are quite strong, with sufficient ablations and additional comparisons to support that the major components of the dataset construction process actually matter for performance.

Strengths
- There are a lot of parts to the dataset construction, but it is explained well – I've listed any of my confusions in Weaknesses and Questions.
- The results in Tables 1 and 3 for continued pretraining and two-stage fine tuning look excellent. Training on TracePile seems to quite generally boost a range of models on a wide range of benchmarks, with few and minor regressions. The further appendix results (Table 7) look in line with this as well. The improvements on benchmarks like CLRS and GraphInstruct that were the basis for parts of the training dataset is not at all surprising, but they also still see quite considerable boosts to the wide array of other benchmarks. The number of datasets used in this evaluation is quite impressive. I appreciate that Table 3 doesn't use the base model but rather the TuluSFT trained model, which does a lot better than the base model on some benchmarks (especially near the bottom of the Table), as this is helpful for disentangling the effect of the second stage of fine tuning.
- The Discussion section with four additional research questions is very well done. In particular I appreciated the ablations of RQ2, which are important to showing that all of the complexity involved (i.e. the numerous sources of data and the diversity features) actually contribute to performance. And similarly RQ3 was important to showing that it's actually worth it to train on such a large dataset.

Weaknesses
- Figure 3 is a little confusing for a few (fixable) reasons.
    - One is that the LLama-3.1-8B line (blue) doesn't make a lot of sense as it's own line, since it doesn't scale with the data and is actually just a single point – and in fact it's just the *zero* data point which could go to the left of 50K. Based on how it is 60.0 in the Math domain, it must be the TuluSFT tuned model (the "Base" model from Table 3) so I believe this would make sense as a Data Scale = 0 point.
    - Secondly, it's confusing that the x axis is nonlinear and has quite variable gaps. Plotting it myself, I think this could look fine with a linear scale from 0 to 4.3M on the x axis. It ends up looking like a rather sharp first jump (in the 0K to 50K segment) then smoothing out.
    - It could make sense to make the very last segment a different color or something, to indicate that this is coming from a different process (rejection sampled data augmentation)
- Details are missing in a handful of places:
    - In RQ4 of the discussion the rejection sampling based augmentation to make TracePile++ is mentioned, but details aren't given – including these in an appendix would be helpful.
    - Lines 159-161 state: "We validate correctness by comparing these outputs to ground-truth traces generated via code instrumentation or symbolic tools, discarding samples that fail consistency checks". This feels key to data quality, and I'd appreciate more details on it (e.g. in an Appendix, but also in the response to this review). Am I correct that this is separate from the intermediate results checks from the Classical Algorithm Code section? Is this applied to many of the problems from the competition code? How does the code instrumentation work, and what symbolic tools are used?

Quick fixes and suggestions
- "CoE formats significantly" (line 139) I think there's a missing word in here?
- Table 2 has a typo in comma placement in the entry "117,083,6"
- The abstract says "TraceMind" instead of "TracePile" in one place.
- line 230 should just say Tables 1 and 3, not Tables 2 and 3
- I found Figure 1 (the splash figure) a bit difficult to read without context – in particular, I think a longer caption could help clear it up. In particular it would be helpful to explain that the 3 boxes on the left are actually 3 different data sources for the dataset. Perhaps putting the size column of each of these from Table 2 into this figure would also be a nice touch.

---

> ### Author Rebuttal · Authors · 2025-07-29
>
> We sincerely thank you for your comprehensive review and the constructive feedback shared. Below are our responses for your insightful comments.
>
> ## Q1: Improvements of Figure 3, including the LLama-3.1-8B line, x-axis scaling, and final segment distinction.
>
> We appreciate the reviewer’s insightful feedback on Figure 3. We will plot the LLama-3.1-8B as a single point at Data Scale = 0, use a linear x-axis from 0 to 4.3M, and visually distinguish the final segment with a different color to indicate rejection-sampled data augmentation. These adjustments will enhance clarity and resolve the concerns raised.
>
> ## Q2: Additional details about rejection sampling for constructing TracePile++.
>
> We thank the reviewer for highlighting this crucial aspect, which directly touches on the diversity and quality of our dataset. Specifically, we construct TracePile++ by sampling five CoE rationales per problem using Qwen-2.5-72B-Instruct, and select semantically diverse yet correct outputs using Sentence-BERT for semantic differentiation. The resulting samples, clearly differing from the original TracePile outputs, are incorporated into TracePile++. We will include comprehensive details of this method in the appendix of the camera-ready version.
>
> ## Q3: Clarification on the correctness validation approach mentioned in lines 159–161, including code instrumentation and symbolic tools.
> The reviewer insightfully identifies a core component of our data quality assurance process. To ensure correctness, particularly for Algorithmic Competition Code, we utilize Python's **snoop** tool, which provides detailed execution traces by capturing variable states and changes throughout code execution. These generated traces are then systematically compared with the model-generated CoE outputs, discarding inconsistent samples. This validation method, distinct from intermediate result checks used in Classical Algorithm Code, ensures semantic fidelity and consistency. We will add detailed explanations of this process, including the use of snoop, in the appendix of the revised manuscript.
>
> ## Q4: Minor corrections and clarity improvements (typos, table references, figure readability).
>
>  We greatly appreciate the reviewer's careful proofreading and thoughtful suggestions to enhance clarity. We acknowledge these minor mistakes and readability issues, and sincerely apologize for the oversight. We will correct the mentioned typos (e.g., "CoE formats significantly", "117,083,6"), replace "TraceMind" with "TracePile" consistently in the abstract, and properly reference Tables 1 and 3 instead of Tables 2 and 3. Furthermore, we will expand the caption of Figure 1, clearly indicate the three distinct data sources, and consider integrating dataset size information to improve readability. All these improvements will be incorporated into the camera-ready version.
> ## Q5: Reasoning behind the selection of benchmarks in Table 4.
>
> The reviewer raises an important point about benchmark selection. The benchmarks shown in Table 4 were specifically chosen to ensure a fair comparison, as the CodeI/O models based on TuluSFT are not publicly available. Thus, we selected benchmarks explicitly evaluated in both the CodeI/O paper and our own study to enable a direct and equitable comparison.
>
> ## Q6: Clarification of the conclusion in lines 306–308 regarding the effects of code rewrites and query diversification.
>
> The reviewer insightfully highlights a key point regarding our interpretation of results. In our analysis, we consider performance on logical reasoning benchmarks as a strong indicator of structural generalization. We observed that removing code rewrites resulted in a more noticeable performance drop on logical reasoning tasks, while removing query diversification had a greater negative impact on mathematical reasoning. This differential pattern guided our conclusion that code rewrites predominantly improve structural generalization, whereas query diversification mainly enhances task-specific interpretability. We will clarify this reasoning explicitly in the revised manuscript.
>
> ## Q7: Clarification on the meaning of "control-flow interpretation" mentioned in lines 154-155.
>
> We thank the reviewer for highlighting this ambiguity. Indeed, the reviewer’s interpretation aligns correctly with our intent. By "control-flow interpretation," we specifically refer to tracking changes in structures like arrays, stacks, or heaps during code execution—commonly involved in classical algorithmic tasks. While such tracking is prevalent in most algorithmic tasks involving loops or conditionals, a substantial portion of our dataset consists of mathematical code, which frequently does not require detailed control-flow tracking. We will explicitly clarify this distinction in the revised manuscript.

---

> > ### Comment · Reviewer_R7E8 · 2025-08-05
> >
> > Thank you for the rebuttal and clarifications! While your other responses were helpful, I'm still a little confused about your reply Q6. *Why* do you consider performing well at logical reasoning to be a strong indicator of "structural generalization", while mathematical reasoning is an indicator of "task-specific interpretability"? Perhaps in this discussion it would be helpful if you spell out what exactly you mean by those terms? Thank you for your time and discussion.

---

> > > ### Author Response · Authors · 2025-08-05
> > > **Rebuttal by Authors**
> > >
> > > Thank you for the follow-up question. Your point is excellent, and we agree that a clear definition of our terminology is crucial.
> > >
> > > Our distinction between "structural generalization" and "task-specific interpretability" is based on the relationship between our training data and the downstream benchmarks.
> > >
> > > - **Task-specific Interpretability (Mathematical and Algorithmic Reasoning)**: We use this term to describe the model's ability to interpret and apply reasoning patterns within a known domain. Our TracePile training corpus already contains mathematical code problems and graph algorithms. When we observed that query diversification—which introduces variations to a problem's presentation—had a strong positive effect on these tasks, we concluded that it primarily enhances the model's ability to better interpret and apply existing, known reasoning patterns to similar problems.
> > >
> > > - **Structural Generalization (Logical Reasoning)**: We use this term to describe the model's ability to apply reasoning principles to entirely new and unseen problem structures. Our training data does not contain examples with the same data format or problem types as logical reasoning benchmarks like Zebra Puzzle or Ruletaker. Therefore, any performance improvement on these benchmarks cannot be a result of direct knowledge transfer. The observed gains must stem from a more abstract, fundamental capability to trace and manipulate logical flow and state, a skill we believe is fostered by the fine-grained, step-by-step nature of our Chain of Execution data. We consider this a form of "structural generalization."
> > >
> > > We will make sure to define these terms explicitly and provide this reasoning in the revised manuscript.

---

> > > > ### Comment · Reviewer_R7E8 · 2025-08-05
> > > >
> > > > Ah, this clarifies things! Apologies, I read the paper a while ago and forgot that the logic domains were brought in as an out-of-domain role, thank you for the reminder.

---

> > > > > ### Author Response · Authors · 2025-08-06
> > > > >
> > > > > Thank you for your response and for taking the time to review our rebuttal. We appreciate your feedback.

---

### Comment · Area_Chair_to99 · 2025-08-05

Dear reviewers,

Thanks for your contribution to the reviews.

The authors have put rebuttal to the reviews, please look at the rebuttal and make responses accordingly. Note that engaging the discussion is important at this phase.

Please make sure that you give necessary comments to the rebuttal and then make acknowlegement, therefore the authors can have a better understanding whether their rebuttal help to your concerns.

Best,
AC

---

### Note · Authors · 2025-08-14

We sincerely thank the reviewers for their diligent feedback and constructive engagement throughout this process. We are grateful for the positive reception of our work.

Reviewers consistently commended our paper for:

- The value and contribution of the TracePile dataset and comprehensive training settings (acknowledged by **all reviewers**).

- Its compelling hypothesis (`Reviewer fNHA`).

- A well-explained and diverse TracePile dataset (`Reviewer R7E8, 2Sy7, fNHA`).

- Thorough evaluation across a wide range of benchmarks (`Reviewer R7E8, ruAL`).

- Strong and generalizable performance boosts achieved by models trained on TracePile (`Reviewer R7E8, 2Sy7`).

- Insightful ablations (`Reviewer R7E8, ruAL`).

- Successful implementation of intermediate result verification (`Reviewer 2Sy7`).

During the rebuttal phase, we directly addressed several key points to further strengthen our submission:

- **Fairness of Baselines**: We clarified training epoch differences and provided comprehensive new comparison tables against various supervision approaches (pure code/math, generic instruction, CodeI/O), showcasing TracePile's superior generalized reasoning.

- **Related Work & Novelty**: We elaborated on our dedicated "Related Works" section in Appendix A and precisely articulated TracePile's novelty in terms of CoE granularity, diverse data sources, and multi-stage training paradigms, including its unique contribution to the CPT stage.

- **Intrinsic Evaluation**: We presented a new experiment demonstrating significantly higher "Intermediate Step Accuracy" (correct output AND all intermediate steps correct), confirming our model learns to generate truly correct traces.

- **Data Statistics & Limit**: We corrected the average token count, clarifying that few-shot prompt examples were inadvertently included, providing a revised average token length consistent with our 8k limit.

- **Generalization Terminology**: We provided further clarification on our definitions of "structural generalization" and "task-specific interpretability" based on the characteristics of our training data and benchmarks.

We are confident that these clarifications and additional analyses significantly strengthen the paper and address all raised concerns. We eagerly anticipate your final decision.

Sincerely,

Anonymous Authors

---

### Decision · Program_Chairs · 2025-09-17

**Decision:**

Accept (poster)

**Comment:**

This paper proposes a chain-of-execution formulation to transform the code to step-by-step evaluation natural language sentences, with such transformation, the created data then can be better leveraged for general reasoning. The authors generate 2.6M data with 25B tokens and demonstrate the effectiveness of the data and the method on multiple domains.

Reviewers agree the paper is well motivated and the method is reasonable and novel. Compared with other methods, the approach shows great improvements. The experimental results deeply impress the reviewers, which shows a general reasoning boost method.
During the rebuttal phase, the authors and reviewers have necessary discussions and the authors have addressed most of the concerns, except one point about the open source data. The authors are encouraged to make necessary revision.